# MIST: Mutual Information Estimation via Supervised Training

**German Gritsai**                                                        *gritsai.gm@gmail.com*
*Université Grenoble Alpes, CNRS, Grenoble INP, LIG*

**Megan Richards**                                                       *megan.richards@nyu.edu*
*New York University*

**Maxime Méloux**                                              *melouxm@univ-grenoble-alpes.fr*
*Université Grenoble Alpes, CNRS, Grenoble INP, LIG*

**Kyunghyun Cho**                                                     *kyunghyun.cho@nyu.edu*
*New York University*

**Maxime Peyrard**                                            *peyrardm@univ-grenoble-alpes.fr*
*Université Grenoble Alpes, CNRS, Grenoble INP, LIG*

**Reviewed on OpenReview:** *https://openreview.net/forum?id=Qi4JgS2PLw*

## Abstract

We propose a fully data-driven approach to designing mutual information (MI) estimators. Since any MI estimator is a function of the observed sample from two random variables, we parameterize this function with a neural network (MIST) and train it end-to-end to predict MI values. Training is performed on a large meta-dataset of 625,000 synthetic joint distributions with known ground-truth MI. To handle variable sample sizes and dimensions, we employ a two-dimensional attention scheme ensuring permutation invariance across input samples. To quantify uncertainty, we optimize a quantile regression loss, enabling the estimator to approximate the sampling distribution of MI rather than return a single point estimate. This research program departs from prior work by taking a fully empirical route, trading universal theoretical guarantees for flexibility and efficiency. Empirically, the learned estimators largely outperform classical baselines across sample sizes and dimensions, including on joint distributions unseen during training. The resulting quantile-based intervals are well-calibrated and more reliable than bootstrap-based confidence intervals, while inference is orders of magnitude faster than existing neural baselines.

## 1 Introduction

Mutual information (MI) is a measure of nonlinear statistical dependence between two random variables, quantifying how much knowing one variable reduces the uncertainty about the other (Shannon, 1948; MacKay, 2003). For two continuous random variables $X$ and $Y$, the MI is written in terms of their joint distribution $P_{(X,Y)}$ and marginal distributions $P_X$ and $P_Y$:

$$I(X;Y) = \int_{\mathcal{Y}} \int_{\mathcal{X}} P_{(X,Y)}(x,y) \log \left( \frac{P_{(X,Y)}(x,y)}{P_X(x) \, P_Y(y)} \right) \, dx \, dy. \tag{1}$$

Mutual information is one of the most widely applied information measures in data science, finding applications in feature selection (Peng et al., 2005; Kwak & Choi, 2002), causality (Butte & Kohane, 1999),

representation learning (Chen et al., 2016; Higgins et al., 2018; Tishby & Zaslavsky, 2015; Zhao et al., 2018), reinforcement learning (Oord et al., 2018; Pathak et al., 2017), and generative modeling (Alemi et al., 2016; Alemi & Fischer, 2018; Huang et al., 2020).

Calculating MI requires access to the joint and marginal distributions, which are rarely known in practice. Consequently, most approaches infer MI from finite samples drawn from these distributions. Existing estimators fall into two main families: density estimators and density ratio estimators. **Density estimators** approximate the underlying densities, for example, using kernel density estimation, $k$-nearest neighbors, or variational autoencoders. MI can then be computed by substituting these estimates into one of MI's definitions (such as Equation 1). Prominent density estimators include KSG (Kraskov et al., 2004) and MINDE (Franzese et al., 2023). **Density ratio estimators** instead estimate the relationship between the joint and marginal distributions, often using variational lower bounds. Notable examples include MINE (Belghazi et al., 2018), NWJ (Nguyen et al., 2010), and SMILE (Song & Ermon, 2019). In both families of approaches, neural methods have been developed to improve the estimator's capacity to capture complex density functions or density relationships.

Broadly, these neural methods have achieved better performance on *moderately* complex distributions and high-MI settings. However, both families of approaches struggle in challenging settings that are common in practice: high dimensionality, limited samples, complex distributions, and high mutual information (Czyż et al., 2023; McAllester & Stratos, 2020). Examples of these settings include applications in genomics (Klein et al., 2015), neuroscience (Ganguli & Sompolinsky, 2012), astronomy (Pandey & Sarkar, 2017), and machine learning (Goldfeld et al., 2018). While previous benchmarking efforts have expanded the set of distribution families considered, MI estimators have rarely been empirically evaluated under these challenging conditions individually, and even more rarely in combination. Most empirical evaluations have studied estimation for fewer than 5 distributions, with a minimum sample size of 10,000 or more, and in higher dimensions primarily for normal distributions (see Appendix A.3). Existing work has instead primarily focused on theoretical guarantees of low bias, consistency with increasing samples, and the tightness of variational bounds (Belghazi et al., 2018; Nguyen et al., 2010; Song & Ermon, 2019).

In this work, we propose a fully data-driven, empirical approach to designing mutual information estimators. Instead of building estimators that compute MI by approximating density functions or their relation, we learn to predict MI directly from data. We parameterize a mutual information estimator as a neural network and train such a network end-to-end on a large meta-dataset of synthetic joint distributions with known ground-truth MI values. Our proposed model, MIST (Mutual Information estimation via Supervised Training), handles arbitrary sample sizes and dimensions, and features built-in uncertainty estimates. This meta-learning approach takes inspiration from amortized inference procedures such as simulation-based inference (SBI; Papamakarios et al., 2019; Cranmer et al., 2020), amortized Bayesian inference (ABI; Gonçalves et al., 2020; Elsemüller et al., 2024; Radev et al., 2023; Avecilla et al., 2022; Gloeckler et al., 2024), and meta-statistical learning (Peyrard & Cho, 2025).

We focus on empirical settings underexplored in previous work but common in practice: fewer samples (10-500), higher dimensions (2-32), and a wider range of MI values (0-40) across 23 distribution types. We find that MIST substantially outperforms existing methods across these conditions, especially in low-sample regimes. We provide a detailed study of sample requirements under increasing dimensionality, finding that MIST requires approximately half as many samples on average to provide reliable MI estimates compared to the next best estimator. The quantile-based intervals are well-calibrated and more reliable than bootstrap alternatives, while inference is orders of magnitude faster than existing neural methods. MIST generalizes effectively to unseen synthetic distributions in our framework, and improves with increasing sample size beyond its training experience. We discuss the limitations of our learning-based approach in Section 5, including the generalization characteristics observed in our experiments. Finally, we provide a training-free mechanism to detect far out-of-distribution settings where MIST's performance may degrade.

We provide an open-source library[1] for training and evaluating meta-learned MI estimators, and include a detailed discussion of opportunities for future work in Section 6.

---

[1] https://github.com/grgera/mist

## 2 Related Work

Mutual information estimation is a foundational challenge across data science tasks, for which a wide variety of estimation methods have been developed. Although these approaches vary in their mathematical foundations and computational strategies, they can be grouped into two main families: density estimators and density ratio estimators. See Appendix A.2 for a detailed discussion of previous estimators and our choices in categorizing them.

**Density Estimators.** Density estimators compute mutual information estimates by performing intermediate approximations of density functions. Histogram MI estimators approximate the joint and marginal density functions in Eq. 1 by partitioning the support of each variable into a set of bins and counting the frequencies in each (Fraser & Swinney, 1986). Histogram estimators suffer from very poor scaling and are highly sensitive to the choice of bins, leading to research on optimizing the bin selection (Darbellay & Vajda, 1999). Kernel density estimators (KDE; Moon et al., 1995; Silverman, 2018) were later developed to help avoid this sensitivity by replacing fixed bins with kernel functions centered at each sample point. Kozachenko (1987) designed a method for entropy estimation using neighborhood distances, an approach which was later refined with weighted variants (Berrett et al., 2019). The Kraskov-Stögbauer-Grassberger estimator (KSG; Kraskov et al., 2004) extended this k-nearest neighbor framework to mutual information estimation using MI's expression in terms of conditional entropy (see Appendix A.1). KSG is one of the most broadly used MI estimators, especially for low-sample settings, and serves as a primary benchmark comparison in this work. Several approaches have explored the use of deep generative models for density estimation. One line of work uses VAEs or normalizing flows to reconstruct the joint and marginal density functions directly, an approach which suffers from high sample requirements and a limited ability to capture complex structure (Song & Ermon, 2019; Duong & Nguyen, 2023). The Barber-Agakov bound (BA; Barber & Agakov, 2004) provides a variational lower bound on MI by introducing a variational approximation $q(x|y)$ to the true conditional $p(x|y)$. Several works have applied this bound using conditional generative models with different choices for the marginal distribution (Alemi et al., 2016; Chalk et al., 2016; Kolchinsky et al., 2019). More recently, MINDE (Franzese et al., 2023) trains conditional or joint diffusion models to learn score functions, then applies the Girsanov theorem to compute KL-divergence as the expected difference between joint and marginal scores. VCE (Chen et al., 2025) is a concurrent work that expresses MI as the negative differential entropy of the vector copula density, which they estimate by first using normalizing flows to model the marginal densities, then applying maximum likelihood estimation to the copula. Another line of density estimators has focused on building MI estimators for complex settings by learning methods to simplify the distributions before applying classical estimators or computing MI directly. MIENF (Butakov et al., 2024) learns a pair of normalizing flows that apply MI-preserving transformations to the joint distribution, enabling closed-form MI calculation. Latent-MI (LMI; Gowri et al., 2024) trains a network to compress data with minimal mutual information loss, then applies KSG to the low-dimensional embeddings. These methods have improved abilities to handle high-dimensional data, but have large sample and computational requirements, and rely on the existence of simple or low-dimensional structure in high-dimensional data.

**Density Ratio Estimators.** Density ratio estimators compute mutual information by estimating the relationship between the joint density $P(X, Y)$ and the product of marginal densities $P_X P_Y$. Since mutual information can be equivalently expressed as the Kullback-Leibler divergence $D_{KL}(P(X, Y) \| P_X P_Y)$ (Donsker & Varadhan, 1975), many density ratio methods leverage variational bounds on KL divergence. Several methods have been developed by defining discriminative tasks that implicitly predict this density ratio. MINE (Belghazi et al., 2018) trains a neural network critic to differentiate joint and marginal samples, optimizing the critic to maximize the Donsker-Varadhan (DV) lower bound. The Nguyen-Wainwright-Jordan estimator (NWJ; Nguyen et al., 2010) trains a critic model similarly, but uses an alternative variational bound which offers lower variance at the cost of a looser bound. InfoNCE (Oord et al., 2018) trains a critic to identify true joint samples among negative samples drawn from the marginals.

SMILE (Song & Ermon, 2019) combines the InfoNCE and NWJ bounds through a clip operation, interpolating between their respective bias-variance profiles to achieve more flexible trade-offs. The limitations of variational estimators are well-studied in McAllester & Stratos (2020), Song & Ermon (2019), and Poole et al. (2019), and include high variance estimates, high sample requirements, and poor estimates for high MI settings. We refer the reader to Poole et al. (2019) for a detailed review of MI variational bounds. FMMI (Butakov et al., 2025) learns a coupling transform to map between the joint distribution and marginal product, using the expected divergence of the velocity field as the MI estimate. InfoNet (Hu et al., 2024) trains a neural network to predict the output of a trained MINE critic, removing the need to train a critic on each new set of samples. Like our work, this approach achieves substantial test-time efficiency gains through large-scale pretraining. However, our works differ in the learning objective: InfoNet learns to emulate an algorithm (MINE), whereas MIST learns an algorithm that allows it to predict the MI directly from samples.

**Machine Learning for Statistical Inference.**  Our work aligns with the broader research direction on learning inference procedures. A relevant related direction concerns neural processes (Garnelo et al., 2018b;a; Kim et al., 2019; Gordon et al., 2020; Markou et al., 2022; Huang et al., 2023; Bruinsma et al., 2023), which can predict latent variables of interest from datasets (Chang et al., 2025) by leveraging transformers (Nguyen & Grover, 2022) and Deep Sets (Zaheer et al., 2017) to enforce permutation invariance (Bloem-Reddy & Teh, 2020). A related approach, known as prior-fitted networks, has demonstrated that transformers can be effectively repurposed for Bayesian inference (Müller et al., 2022) and optimization tasks (Müller et al., 2023). Additionally, there is growing interest in using trained models to assist in statistical inference tasks (Angelopoulos et al., 2023) and optimization (Lueckmann et al., 2017; Liu et al., 2020; Simpson et al., 2021; Amos, 2023) like simulation-based inference (SBI; Papamakarios et al., 2019; Cranmer et al., 2020), amortized Bayesian inference (ABI; Gonçalves et al., 2020; Elsemüller et al., 2024; Radev et al., 2023; Avecilla et al., 2022; Gloeckler et al., 2024) and the learning of frequentist statistical estimators (Peyrard & Cho, 2025).

## 3   Learning Framework

The methods available today decompose MI estimation into intermediate tasks of density function or density ratio estimation. To estimate a density function (or ratio) beyond simple settings, existing methods require large amounts of samples as part of inference. Even with access to a high number of samples, density estimation itself is a notoriously difficult and unsolved problem, particularly for high-dimensional or complex distributions (Silverman, 2018; Mészáros et al., 2024). As a result, the MI estimators available today still frequently fail in many realistic data settings. They scale poorly to high-dimensional data and complex distributions, are unstable for low-sample settings, and require expensive inference procedures. An additional consequence is that studying and improving MI estimators is challenging, with new methods often resulting from theoretical discoveries.

Our work offers an alternative philosophy for designing mutual information estimators. Instead of building estimators that compute MI with a pre-specified algorithm, such as approximating density functions or their ratios, we propose to predict MI directly from data. We take a fully empirical perspective, asking: *Can a model learn a generalizable algorithm for predicting mutual information directly from samples?* To learn this algorithm, we parameterize a mutual information estimator as a neural network that we call MIST (Mutual Information estimation via Supervised Training). MIST takes as input a set of $n$ paired samples $\{(x_i, y_i)\}_{i=1}^n$ drawn from a joint distribution $P_{XY}$, and outputs an estimate of $I(X; Y)$. The model is trained end-to-end on a large meta-dataset of synthetic joint distributions with known ground-truth MI values. We provide a visualization of our framework in Figure 1 and describe the meta-dataset design, architecture, and training choices below.

Figure 1: We propose a fully data-driven, empirical approach to designing mutual information estimators. We design a large empirical meta-dataset composed of samples from a set of distributions with known MI (left), and train a SetTransformer-based model to predict MI directly from sets of samples (right).

**The Learning Task**   Let $\Gamma$ denote a family of joint data-generating distributions over $\mathcal{X} \times \mathcal{Y}$. For any $\gamma \in \Gamma$, let $I_\gamma$ denote the mutual information between $X$ and $Y$ under $\gamma$. A mutual information estimator observes a finite dataset $D := \{(x_i, y_i)\}_{i=1}^n$ drawn i.i.d. from $\gamma$, and aims to estimate $I_\gamma$ from samples alone. We parametrize the estimator as a function $f_\theta : \bigcup_{n=1}^\infty (\mathcal{X} \times \mathcal{Y})^n \to \mathbb{R}_{\geq 0}$, where $\theta$ are neural network weights. To train $f_\theta$, we assume access to a *meta-distribution* $P_\Gamma$ over generative laws. First, a distribution $\gamma \sim P_\Gamma$ is sampled, then a dataset $D$ is drawn from $\gamma$, together with its ground-truth mutual information $I_\gamma$. The training objective is to minimize the mean squared error:

$$\mathcal{L}_{\text{MSE}}(\theta) = \mathbb{E}\Big[ \big(f_\theta(D) - I_\gamma\big)^2 \Big], \tag{2}$$

where the expectation is taken over $D \sim \gamma$ and $\gamma \sim P_\Gamma$. In the limit of sufficient model capacity and perfect optimization, $f_\theta$ converges to the *Bayes-optimal regression function*: $f^*(D) = \mathbb{E}[\, I_\gamma \,|\, D \,]$. That is, the optimal estimator outputs the posterior expectation of the mutual information conditioned on the observed dataset. This shows the soundness of the approach, which is further clarified by the following decomposition of the MSE loss:

$$\mathcal{L}_{\text{MSE}}(\theta) = \mathbb{E}\Big[\big(f_\theta(D) - f^*(D)\big)^2\Big] + \mathbb{E}[\text{Var}(I_\gamma \,|\, D)], \tag{3}$$

The second term corresponds to the irreducible epistemic uncertainty due to the fact that, for a finite $n$, the dataset $D$ does not uniquely identify the generative law $\gamma$. The irreducible error in MSE comes from the difficulty of the statistical inference task.

**Built-in Uncertainty Quantification**   Uncertainty quantification is a desirable feature for MI estimators, allowing practitioners to assess reliability and construct confidence intervals. However, existing methods lack built-in uncertainty estimates. The main available approach is bootstrapping, which is prohibitively expensive in practice, as inference on each bootstrap sample requires refitting a density or density ratio model. Our meta-learning paradigm offers an opportunity to incorporate uncertainty estimation into model design by modifying the loss. In addition to our MSE-trained model (MIST), which predicts a point estimate of MI, we also train a quantile regression model ($\text{MIST}_{\text{QR}}$) to predict the $\tau$-quantile of the sampling distribution for any given $\tau \in [0, 1]$. We use the pinball loss (Steinwart & Christmann, 2011), defined for a a target quantity $q$ and predicted quantity $\hat{q}$ as:

$$L_\tau(q, \hat{q}) = \max(\tau(q - \hat{q}), (1 - \tau)(\hat{q} - q)). \tag{4}$$

By querying multiple values of $\tau$, we can approximate the full sampling distribution of MI. Furthermore, these estimates are obtained as the result of a forward pass and are therefore cheap to compute. We evaluate the calibration and reliability of these quantile-based intervals against bootstrap alternatives in Section 4.4 and include both model variants throughout our experiments.

**Meta-Dataset**   Training our meta-learned estimators in a supervised manner requires distributions with known MI. We generate these distributions using BMI (Czyż et al., 2023), a library designed to create complex benchmarks for MI estimation. BMI exploits the invariance of mutual information to homeomorphisms, applying invertible transformations to simple base distributions to produce more complex and challenging distributions. We partition the base distribution families into separate train and test sets, that is, the meta-distribution used for training has a different support than the one used for testing. For each base distribution, we apply BMI's suite of invertible transformations and generate variants across dimensions ranging from 2 to 32, yielding two distribution sets $\Gamma_{\text{train}}$ and $\Gamma_{\text{test}}$. From these distributions, we construct a training meta-dataset $\mathcal{M}_{\text{train}}$ and two test meta-datasets differing only in scale: $\mathcal{M}_{\text{test}}$, and $\mathcal{M}_{\text{test-extended}}$. Both scales of test meta-datasets contain an in-meta-distribution (IMD) subset, in which the distribution family was seen in $\Gamma_{\text{train}}$, and an out-of-meta-distribution (OoMD) subset. Each meta-datapoint in our meta-datasets consists of $n$ paired samples $\{(x_i, y_i)\}_{i=1}^n$ drawn from a joint distribution $\gamma \in \Gamma$, along with the known mutual information $I_\gamma(X; Y)$ as the label. The value of $n$ varies between 10 and 500 across each of our train and test meta-datasets.

The resulting training meta-dataset $\mathcal{M}_{\text{train}}$ contains 16 base distribution families and 625,000 meta-datapoints. $\mathcal{M}_{\text{test-extended}}$ contains 806,000 meta-datapoints from 13 distribution families, of which 6 are shared with $\mathcal{M}_{\text{train}}$. Meanwhile, $\mathcal{M}_{\text{test}}$ has a reduced size of 2,340 meta-datapoints, created to allow the testing of slower baselines that would take prohibitively long to run on the extended test set. The test sets are sampled to uniformly cover all sample sizes, dimensions, and base distribution families. The contents and generation of the meta-datasets are described in more detail in Appendix B.

**Model Architecture**   Building an estimator using our framework presents three main architectural challenges. First, the model must process datasets of varying size while remaining invariant to sample ordering. The Transformer architecture is naturally permutation-invariant and has been a common choice to ingest datasets as inputs (Xu et al., 2024). The SetTransformer (Lee et al., 2019) architecture introduced the ISAB block to reduce the quadratic cost of attention by performing attention on a fixed number of learned inducing points. Finally, the SetTransformer++ architecture (Zhang et al., 2022) further improves upon the original SetTransformer by introducing set normalization and additional residual connections that enable more stable training with deeper models. We therefore use SetTransformer++ as the basis of our architecture. Second, our estimator must process samples of varying dimensions. To address this, we introduce an additional attention block that operates over the dimension axis; this time, permutation invariance is not required. This row-wide attention also operates with a learned fixed inducing point, which then serves as a pooling mechanism into a fixed row dimension. Third, unlike many supervised learning tasks, mutual information is an unbounded measure. This is a long-standing challenge for MI estimators, which typically underestimate MI and fail in high-MI settings. We explored several normalization and multi-task prediction strategies to address this problem when designing our model, and provide our results in Appendix C.1. We found that the simplest solution of directly predicting MI produced the best estimators, provided the meta-dataset includes a sufficient range of MI labels.

Table 1: MIST estimators substantially outperform existing methods across sample size ($n$) and distributions, including for BMI-generated distributions unseen during training (OoMD, right). Shown are the averaged MSE loss values (mean $\pm$ stddev) of each estimator, grouped by the number of samples $n$ given to the estimator, and using 100 bootstrap samples for the standard deviation.

| $n \in$ | In-Meta-Distribution (IMD) | | | Out-of-Meta-Distribution (OoMD) | | |
|---|---|---|---|---|---|---|
| | $[10, 100]$ | $[100, 300]$ | $[300, 500]$ | $[10, 100]$ | $[100, 300]$ | $[300, 500]$ |
| CCA | $4.6e^3$ $\pm 7.4e^3$ | $2.3e^1$ $\pm 7.1e^1$ | $6.7e^0$ $\pm 3.7e^1$ | $4.0e^3$ $\pm 6.8e^3$ | $1.8e^1$ $\pm 7.3e^1$ | $1.2e^1$ $\pm 6.2e^1$ |
| KSG | $3.0e^1$ $\pm 8.7e^1$ | $2.8e^1$ $\pm 1.0e^2$ | $2.7e^1$ $\pm 9.5e^1$ | $3.7e^1$ $\pm 7.7e^1$ | $3.5e^1$ $\pm 8.6e^1$ | $4.3e^1$ $\pm 1.1e^2$ |
| MINE | $6.3e^3$ $\pm 3.7e^4$ | $1.2e^4$ $\pm 1.2e^5$ | $1.2e^5$ $\pm 2.0e^6$ | $7.1e^3$ $\pm 4.1e^4$ | $1.5e^4$ $\pm 2.3e^5$ | $3.3e^4$ $\pm 4.0e^5$ |
| InfoNCE | $1.4e^2$ $\pm 2.5e^2$ | $3.5e^1$ $\pm 1.2e^2$ | $3.2e^1$ $\pm 1.1e^2$ | $3.2e^2$ $\pm 2.1e^3$ | $4.2e^1$ $\pm 9.7e^1$ | $5.0e^1$ $\pm 1.2e^2$ |
| InfoNet** | $3.2e^1$ $\pm 9.3e^1$ | $3.5e^1$ $\pm 1.2e^2$ | $3.7e^1$ $\pm 1.2e^2$ | $3.9e^1$ $\pm 8.2e^1$ | $4.2e^1$ $\pm 9.9e^1$ | $5.5e^1$ $\pm 1.3e^2$ |
| FMMI | $2.8e^1$ $\pm 8.4e^1$ | $2.5e^1$ $\pm 9.8e^1$ | $1.7e^1$ $\pm 6.5e^1$ | $3.3e^1$ $\pm 7.4e^1$ | $2.8e^1$ $\pm 7.7e^1$ | $2.2e^1$ $\pm 6.7e^1$ |
| VCE | $1.5e^1$ $\pm 3.9e^1$* | $1.2e^1$ $\pm 5.1e^1$ | $1.6e^1$ $\pm 3.6e^1$ | $2.6e^1$ $\pm 6.5e^1$* | $2.4e^1$ $\pm 6.6e^1$ | $2.4e^1$ $\pm 5.9e^1$ |
| DV | $3.0e^{19}$ $\pm 2.1e^{20}$ | $1.1e^9$ $\pm 2.2e^{10}$ | $8.5e^6$ $\pm 6.7e^7$ | $1.2e^{19}$ $\pm 6.3e^{19}$ | $8.7e^6$ $\pm 5.1e^7$ | $6.1e^6$ $\pm 5.8e^7$ |
| LMI | $3.2e^1$ $\pm 9.1e^1$ | $3.1e^1$ $\pm 1.1e^2$ | $3.0e^1$ $\pm 1.1e^2$ | $3.8e^1$ $\pm 8.0e^1$ | $3.6e^1$ $\pm 9.1e^1$ | $4.6e^1$ $\pm 1.2e^2$ |
| MINDE | $3.5e^1$ $\pm 9.7e^1$ | $3.2e^1$ $\pm 1.2e^2$ | $2.5e^1$ $\pm 9.2e^1$ | $4.2e^1$ $\pm 8.5e^1$ | $3.8e^1$ $\pm 9.5e^1$ | $3.6e^1$ $\pm 1.0e^2$ |
| MIST | $\mathbf{3.1e^0 \pm 1.0e^1}$ | $3.4e^{-1}$ $\pm 9.1e^{-1}$ | $2.8e^{-1}$ $\pm 9.9e^{-1}$ | $\mathbf{8.0e^0 \pm 1.4e^1}$ | $\mathbf{7.3e^0 \pm 1.9e^1}$ | $\mathbf{8.8e^0 \pm 2.2e^1}$ |
| MIST$_{QR}$ | $\mathbf{3.1e^0 \pm 1.2e^1}$ | $\mathbf{3.1e^{-1} \pm 1.5e^0}$ | $\mathbf{1.2e^{-1} \pm 3.0e^{-1}}$ | $9.5e^0$ $\pm 1.7e^1$ | $9.9e^0$ $\pm 2.5e^1$ | $1.0e^1$ $\pm 2.5e^1$ |

\* VCE fails in the low-sample, high-dimension setting due to the estimated covariance matrix not being definite positive/invertible. This occurs for 99/216 (118/252) points of the IMD (OoMD) subset.

\*\* InfoNet was trained solely on Gaussian mixtures. For a fairer comparison, we report in Appendix C.3 scores computed only on the multivariate Gaussian subset of $\mathcal{M}_{\text{test}}$.

## 4 Experiments

Our experiments demonstrate that MIST outperforms existing methods across sample sizes and dimensions, particularly in low-sample regimes (4.1). MIST exhibits lower bias, lower variance, and higher CI coverage with increasing dimensions and samples (4.2). On average, MIST requires half the samples of the best baseline for reliable estimates in higher dimensions (4.3). The quantile-based intervals are well-calibrated and more reliable than bootstrap alternatives (4.4), while being orders of magnitude faster than existing neural methods (4.6). MIST generalizes to unseen distributions in our framework and sample sizes beyond its training experience (4.5), and benefits from training on diverse dimensions and dataset sizes (4.7).

### 4.1 MIST Outperforms Existing Estimators, Even on Unseen BMI Distributions

We begin by benchmarking our estimators against existing methods on a diverse set of synthetic distributions. In Table 1, we present the average MSE loss of each estimator on $\mathcal{M}_{\text{test}}$, grouped by ranges of meta-datapoint size for clarity. We use the smaller test set of 2,340 meta-datapoints to evaluate all baselines. In this challenging regime of low sample sizes, varying dimensionality, and non-Gaussian distributions, recent estimators struggle to outperform the classical KSG baseline. This aligns with recent findings demonstrating that evaluation on more complex settings reveals weaknesses in modern estimators (Czyż et al., 2023; Lee & Rhee, 2024). Our learned estimators substantially outperform all baselines. Relative to KSG (the next strongest method), our models achieve approximately $10\times$ lower error on distributions seen during training (IMD), and $\sim 5\times$ lower error on unseen distributions (OoMD) generated from BMI. A breakdown by distribution family is provided in Appendix C.5, while median results are provided in Appendix C.2. For subsequent large-scale evaluation, we focus on KSG as the baseline, since it is both computationally feasible on the extended test set and remains the strongest baseline estimator in Table 1, particularly in low-sample regimes.

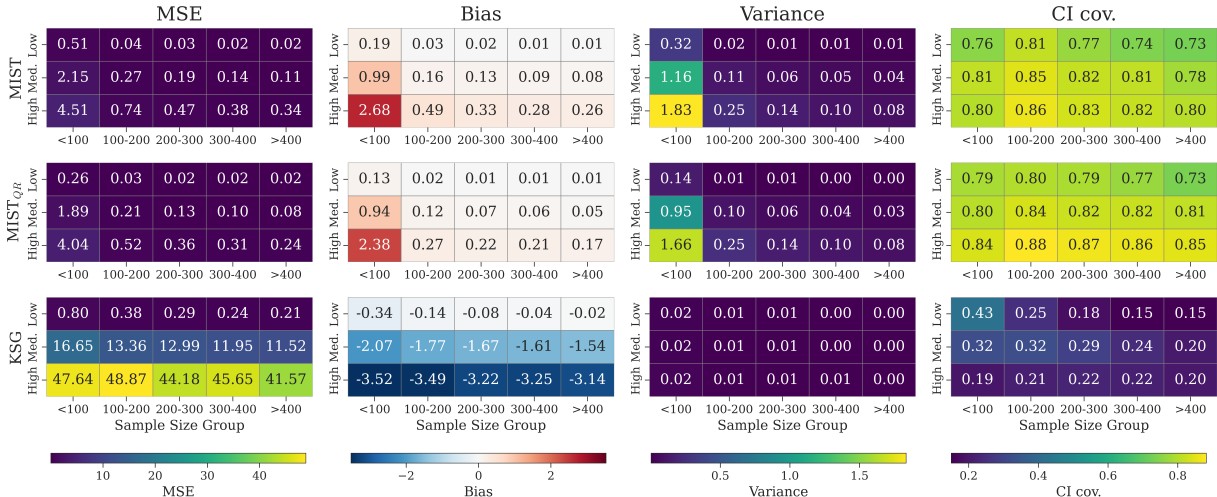

Figure 2: MIST estimators widely outperform KSG, remaining substantially less biased and lower variance across dimensions groups and sample sizes in $\mathcal{M}_{\text{test-extended}}$. CI coverage denotes the fraction of samples per group for which the true MI lies within the 95% bootstrap confidence interval.

## 4.2 MIST Exhibits Lower Bias and Variance Across Settings

Figure 2 further decomposes performance into MSE, bias, variance, and confidence interval coverage for our models (MIST, MIST$_{QR}$) and for KSG. For all estimators except MIST$_{QR}$, confidence intervals are obtained via bootstrap resampling of the input dataset; for MIST$_{QR}$, we use the predicted sampling distribution from the quantile outputs. Each heatmap summarizes average performance as a function of sample size (x-axis) and input dimensionality (y-axis).

Our meta-learned estimators achieve substantial improvements in challenging regimes, with up to 100× lower loss for high-dimensional and low-sample settings. They are nearly unbiased in all $\mathcal{M}_{\text{test-extended}}$ settings, only exhibiting a positive bias for the high-dimensional, low-sample setting. In contrast, KSG exhibits negative bias across all medium- and high-dimensional settings. Notably, our estimators avoid the underestimation problem that plagues existing methods in high dimensions. For variance, our estimators match KSG's low variance for sample sizes above 100. Finally, our estimator's confidence intervals achieve approximately 2× better coverage than KSG across all settings. One of the most well-documented failure modes of existing MI estimators is their negative bias, in which estimators substantially underestimate predictions for distributions with large MI. In Figure 3, we plot the predictions of our estimators in terms of increasing MI. While KSG's estimates plateau quickly, our estimates closely follow the true MI values.

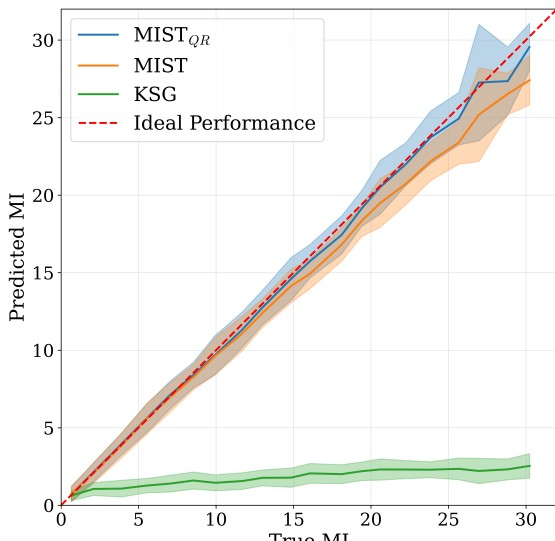

Figure 3: Predicted MI as a function of the true MI on $\mathcal{M}_{\text{test-extended}}$. For each estimator, predictions are aggregated into bins of true MI values.

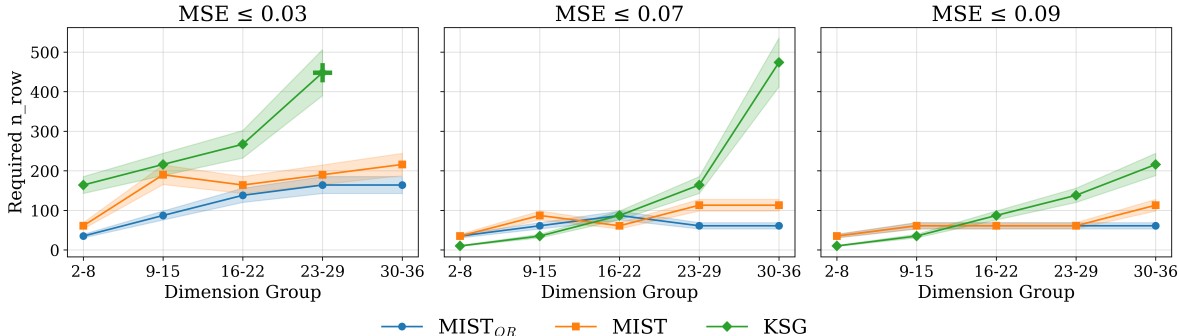

Figure 4: The MIST and MIST$_{\mathrm{QR}}$ models scale substantially better to higher dimensions, requiring roughly half as many samples as KSG to achieve an MSE below the selected thresholds. The "+" marker indicates that over 500 samples ($n_{\mathrm{row}}$) are required to obtain accurate estimates in all higher-dimensional settings.

### 4.3 MIST Requires Fewer Samples to Scale to Higher Dimensions

Traditional evaluations of MI estimators have been constrained to relatively simple distributions and high-sample regimes. While recent efforts have enabled more comprehensive studies on complex distributions, systematic empirical analysis of scaling behavior has remained limited, largely due to the computational expense of evaluating existing methods. Leveraging our larger test set $\mathcal{M}_{\mathrm{test\text{-}extended}}$, we conduct a detailed empirical study of estimator performance across a wide range of dimensions and sample sizes. This enables us to characterize performance frontiers in terms of data requirements. This analysis addresses an important question for practitioners: *How many samples are needed for a reliable estimate?* Figure 4 shows the data requirement profiles of our estimators compared to the KSG estimator. The x-axis represents the dimensionality of the distribution. The y-axis indicates the sample size required to achieve target MSE thresholds of 0.03, 0.07, and 0.09, selected based on our prior heatmap analysis. Our estimators consistently attain the desired accuracy with substantially fewer samples than KSG. In particular, for certain MSE targets, even a sample size of approximately 500 is insufficient for KSG to achieve reliable performance. A complete heatmap of these results is provided in Appendix C.7.

### 4.4 MIST Quantile-Based Intervals are More Reliable Than Bootstrapping

Uncertainty estimates are essential for MI estimation in practice, allowing practitioners to construct reliable confidence intervals and make informed decisions for use based on the uncertainty of the estimation. We evaluate uncertainty quantification for both variants of our estimator, shown in Figure 5 and described below.

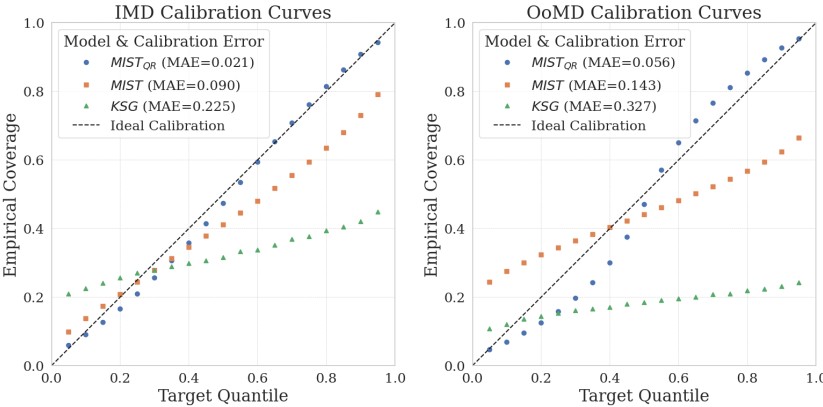

Figure 5: Calibration of the proposed models on $\mathcal{M}_{\mathrm{test,IMD}}$ (left) and $\mathcal{M}_{\mathrm{test,OoMD}}$ (right), computed directly from quantiles for MIST$_{\mathrm{QR}}$ and through bootstrap resampling for MIST and KSG.

Figure 6: MIST models generalize effectively to unseen dimensions and sample sizes. We train MIST and MIST$_{QR}$ on only dimensions up to 16 and fewer than 300 samples (bottom left quadrant), and evaluate their performance on higher dimensions (bottom right), and with more samples (top left), and both (top right). Average MSE is reported for IMD and OMD sets of $\mathcal{M}_{\text{test-extended}}$.

For MIST, we compute confidence intervals via bootstrapping, which is feasible due to our method's fast inference. For the quantile regression model MIST$_{QR}$, we obtain uncertainty estimates by querying the model at different values of $\tau$, directly approximating the sampling distribution. To assess calibration, we compare predicted quantiles against empirical coverage—the proportion of true MI values that fall below each predicted quantile. Figure 5 plots this comparison for both in-meta-distribution and out-of-meta-distribution examples from $\mathcal{M}_{\text{test}}$. Both of our estimator variants substantially outperform KSG in calibration, even on unseen distribution families generated by BMI. This confirms the results reported in Figure 2 showing that confidence intervals built from MIST models often contains the true MI value (for around 80% of instances). The quantile-loss variant MIST$_{QR}$ achieves nearly perfect calibration on distributions seen in training and remains well-calibrated for unseen distributions. Crucially, we obtain these uncertainty estimates at minimal computational cost: computing confidence intervals requires only additional forward passes for either model.

## 4.5 MIST Generalizes to Unseen BMI Distributions, Dimensions, and Sample Sizes

The previous sections demonstrate the unique ability of meta-learned estimators to generalize to unseen BMI distribution families (OoMD) at test-time. The strength of these results motivates a bolder question: *if meta-learned MI algorithms can generalize to unseen distributions, can they also generalize to dimensions and sample sizes beyond their training experience?*

We conduct a controlled study by training models on only the subset of $M_{\text{train}}$ with $D < 16$ and $n < 300$, then evaluating on the full test set $\mathcal{M}_{\text{test-extended}}$ which includes unseen higher dimensions ($D > 16$), unseen larger sample sizes ($n > 300$), and unseen new BMI distributions (OoMD). Figure 6 presents loss heatmaps for in-distribution (IMD) and out-of-distribution (OoMD) test sets. We find that our estimators generalize well to new dimensions and sample sizes for distributions that are in the training meta-distribution (IMD heatmap). Similarly, we find that MIST models generalize well to new distributions when the data dimensionality is similar to that during training (OoMD heatmap, left half). However, they struggle to generalize to unseen distributions when the data also has higher dimensionality than any seen during training (OoMD heatmap, right half). To help assess when MIST predictions are reliable, we introduce a simple approach to estimate how far a test dataset lies from the meta-training distribution. An overview is given in Section 5 and full details are provided in Appendix D.

### 4.6   MIST Achieves Large Gains in Inference Efficiency

Unlike traditional methods that require fitting density or density ratio models for each new dataset, our learned estimators perform inference through a single forward pass. This makes our estimators dramatically more efficient at inference, as seen in Figure 7: 1.7× faster than the most efficient baseline and 4–80× faster than KSG, the best-performing traditional estimator. Training MIST and $\text{MIST}_{\text{QR}}$ each required 3 hours and 45 minutes using a batch size of 256 on an Nvidia A100 GPU. Critically, this one-time training cost is amortized across all subsequent inference tasks.

The combination of fast inference and efficient uncertainty quantification (as discussed in Section 4.4) makes our approach practical for large-scale applications where repeated MI estimation is required, such as hyperparameter tuning, feature selection, or iterative model training (see Section 4.8 for an example of use in feature selection). We estimated the inference time of each baseline on the entirety of $\mathcal{M}_{\text{test-extended}}$ and report the results in Appendix C.6. The high computational costs ($> 1$ week) of evaluating all baselines except for CCA and KSG on $\mathcal{M}_{\text{test-extended}}$, along with the poor performance measured for CCA on the $\mathcal{M}_{\text{test}}$ for low sample sizes, further motivate our selection of KSG as the only baseline for other experiments in this work. On our multidimensional test set, InfoNet's slicing overhead offsets its efficiency gains over optimized methods like MINE.

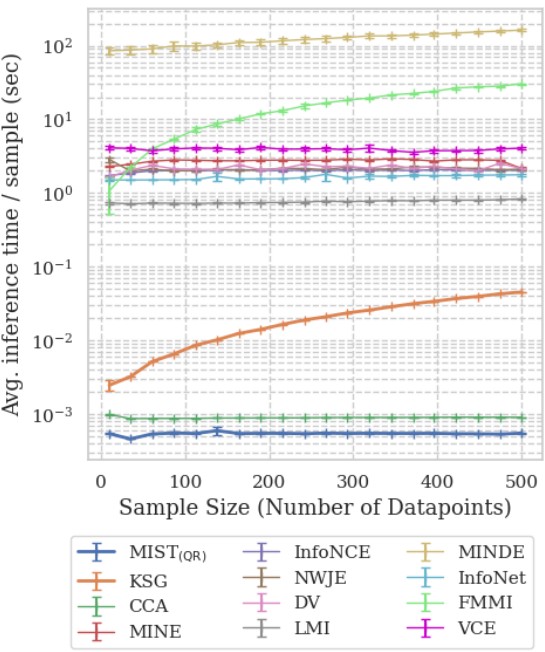

Figure 7: Average inference time for each method with increasing sample size.

### 4.7   Training on Multiple Dimensions Benefits High-Dimensional Estimation

In contrast to existing paradigms, MIST models learn over a diverse meta-distribution rather than fitting an individual dataset of interest. This approach opens many relevant empirical questions regarding the design of the meta-distribution. One of the most prominent is the choice to train dimension-specific models or train one model on mixed dimensionality data. While training over many dimensions has the benefit of generalized use, practitioners may know the dimension of their data a priori.

To study this question, we trained separate (*specialized*) MIST models on subsets of $\mathcal{M}_{\text{train}}$ filtered to contain only meta-datapoints with a fixed input dimension (e.g., 4, 8, 16 or 32). This filtering procedure reduces the number of training data points substantially (by a factor of 31). To provide a comparison with the same number of samples, we also trained additional (*reduced data*) MIST models on a uniformly down-sampled version of $\mathcal{M}_{\text{train}}$ to match the number of samples seen by the individual dimension models. By comparing the base MIST (full training set) with the *specialized* models (only one dimension), we can measure the impact of scaling both the dimensions and dataset size. By comparing the *reduced data* model (downsampled train set) and the *specialized* models (only one dimension), we can isolate the impact of training on multiple dimensions, without increasing dataset scale. In Figure 8, we provide these comparisons for each input dimension in $\mathcal{M}_{\text{test-extended}}$.

We find that scaling both the dimensionality and dataset size (*base* vs *specialized*) substantially improves model performance for high-dimensionality data (2-3× lower MSE for $D \geq 16$) but does not improve performance for low dimensions $D \leq 8$. We hypothesize that as the dimensionality increases, models are more likely to benefit from the regularization strategies learned from a larger and more diverse training distribution. When controlling for the number of samples (*reduced data* vs *specialized*), we find that learning from a mix of data dimensions has lower performance than a specialized model trained on one dimension. We

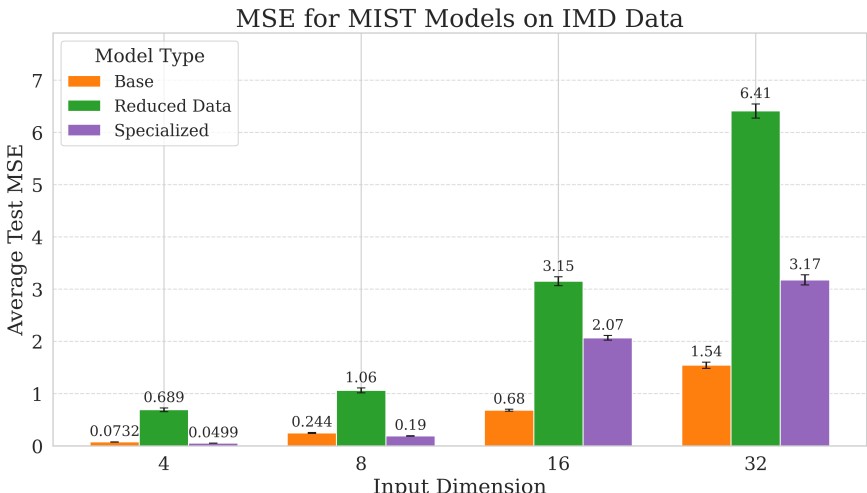

Figure 8: Average MSE obtained by variable- and fixed-dimensionality versions of MIST on subsets of $\mathcal{M}_{\text{test-extended,IMD}}$ with fixed dimensions (x-axis). Error bars represent the standard error $\sigma/\sqrt{n}$. Similar experiments were performed on $\text{MIST}_{\text{QR}}$ and OoMD data, with corresponding plots given in Appendix C.4.

hypothesize that learning across dimensions requires a larger number of samples and longer training times to learn the appropriate regularization strategies required for generalization.

Overall, we find that learning from a large and diverse set of distributions (which can be generated synthetically) is the strongest path to designing effective MIST estimators. Additional results in Appendix C.4 show that the same trend holds for $\text{MIST}_{\text{QR}}$ and in the OoMD setting, the key difference being that the reduced data models' performance does not degrade as much in the OoMD setting. Furthermore, the increased meta-dataset size of the full model further improves its performance by a factor of ∼5 (∼10 for dimension 4) compared to that of the reduced data model.

### 4.8 Feature Selection

We explore two synthetic feature selection settings. In the first, features are scored individually by their mutual information with an outcome variable (*Direct Feature Selection*). In the second, continuous feature weights are learned via gradient ascent on the total mutual information (*Learned Feature Selection*). Together, these settings allow us to study performance both as stand-alone estimators and as optimization objectives. We define a feature matrix $X \in \mathbb{R}^{n \times d}$ as a random normal matrix with $d$ features and $n$ samples. An outcome variable $Y \in \mathbb{R}^n$ is generated by applying a binary weight vector $w^* \in \{0,1\}^{d \times 1}$ where $k$ weights are randomly selected to be 1 (correlated with $Y$), and the others are set to 0. The goal in this task is to select the true informative features, assigning high values for the weights of the $k$ informative features and low values to the uninformative features. We set $d$ to be 32 and $k$ to 2. See Appendix F for further experimental details and example learning trajectories.

For *Direct Feature Selection*, we estimate $MI(X_i, Y)$ for each feature $X_i$ individually, and use these values to select features. Figure 9 shows ROC curves with varying sample sizes in the top row. We find that MIST outperforms KSG and MINE across sample sizes, with largest gains in the lowest sample settings. For *Learned Feature Selection,* Rather than scoring each feature individually, we take a more challenging setting of learning a continuous set of feature weights $w'$ by performing gradient ascent on the collective information. For a candidate set of feature weights $w'$, the MI is calculated on the weighted sum of features $MI(Xw, Y)$. For KSG, we use coordinate descent for optimization, using 100 epochs. For MINE, we trained a new estimator for 200 steps to perform each estimation. We also evaluated a version of MINE that was continually learned, but found this had worse performance (see Appendix F). We plot the ROC curves in Figure 9, and find that MIST-guided learning was more accurate, lower-variance, and more stable across a range of sample sizes and noise levels.

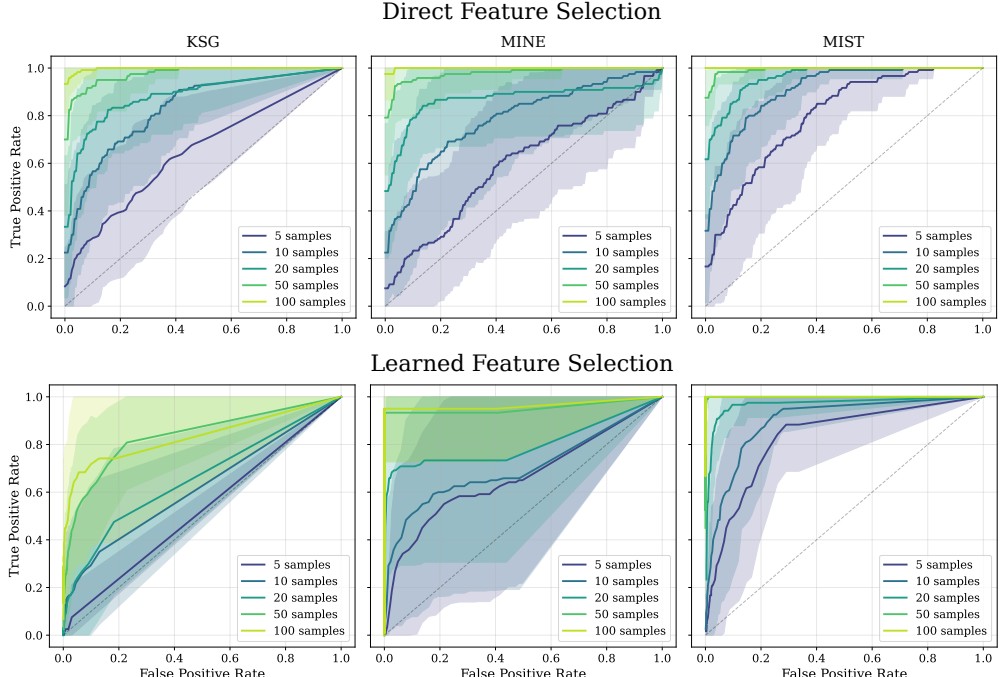

Figure 9: ROC curves of Direct and Learned feature selection with KSG, MINE and MIST Estimates. Results are computed with 3 noise levels and 20 random seeds, with 1 standard deviation shaded. MIST outperforms baselines in both settings.

## 5 Limitations

### 5.1 Theoretical Guarantees

Our estimator's behavior is learned, rather than designed, and therefore does not carry the theoretical consistency guarantees of classical MI estimators. For example, estimator's consistency, the fact that estimate converges to the true MI as $n \to \infty$, is only verified empirically. We studied our estimator's behavior on self-consistency tests from Song & Ermon (2019) in Appendix E, and find that while our estimators are approximately self-consistent under the independence test, they are not self-consistent and produce high variance prediction ratios for the data processing and additivity tests. These properties are motivated by optimization applications of mutual information, and MIST's inconsistency in this test indicates that future work may be required to make MIST reliable as a general-purpose learning objective. We note that reliability of MI estimators during optimization remains an open challenge and a rich area of future work. A learning-based MI framework offers some areas of future promise, particularly in computational efficiency. A learning framework also offers the potential for for designing models, datasets, and losses to induce desired behavior characteristics. But is important to recognize that our estimators do not carry these properties innately and empirical evaluation is required. Moreover, learned estimators are not inherently interpretable and may result in unforeseen failure modes. These limitations parallel the general challenges faced by machine learning models in other domains, highlighting the importance of careful validation, uncertainty quantification, and benchmarking.

### 5.2 Generalization Out of Meta-Distribution

By embracing an empirical learning framework, we inherit the risks of statistical learning. The learned estimator's performance depends on the diversity and coverage of the meta-training distributions, and its generalization to distant out-of-distribution families cannot be guaranteed a priori. In such regimes, performance degradation may affect both point predictions and the confidence intervals produced by MIST$_{QR}$.

Below, we summarize the main generalization trade-offs observed in our experiments, and mitigation strategies.

**Larger Sample Sizes**: MIST generalizes smoothly to larger, unseen sample sizes, and in fact becomes a better estimator (lower MSE) as the number of samples increases, consistent with the behavior of a consistent estimator. In this regime, classical baselines also improve and can partially catch up with MIST; however, MIST remains orders of magnitude faster.

**Unseen Distributions:** MIST exhibits a moderate performance degradation on unseen distribution families still generated with BMI. Despite this drop, it consistently outperforms all baselines, particularly in the low-sample regime, and remains superior for downstream tasks such as feature selection. Our experimental framework evaluates MI estimators on families of distributions that are unseen during training (OoMD), but these distributions are still generated by BMI and do not represent and do not represent arbitrary or adversarial distribution shifts that may be encountered in downstream tasks and challenging for MIST (e.g., discrete distributions or highly structured data).

**Unseen Dimensionalities:** A more challenging scenario arises when generalizing to unseen dimensionalities, especially when combined with distribution shift. In this regime, MIST can still outperform baselines in low sample regime, but suffers from a larger degradation, which also affects the coverage of confidence intervals.

**Larger MI Values** Another important consequence of MIST being a learned estimator is that the range of MI values predicted by MIST are limited to those seen during training. While MIST's training distribution covers a large range of values, it is important for practitioners to consider this limitation in applying MIST.

**Calibration** $\text{MIST}_{QR}$'s calibration is also dependent on the training distribution. In Appendix E, we evaluate $\text{MIST}_{QR}$'s monotonicity and bound consistency in IMD and OoMD settings. We find that $\text{MIST}_{QR}$ achieves monotonic predictions reliably both IMD and OoMD, with breaks primarily for very low MI values ($<0.05$). Both upper and lower bound failures increase significantly out of distribution (lower bound failures from 9.6% IMD to 15.8% OoMD, and upper bound failures 4.3% to 27.5%). In Appendix E, we provide a more detailed explanation of what $\text{MIST}_{QR}$'s quantiles capture, and under what conditions they fail as a bound. This limitation reflects common challenges in many machine learning tasks, where re-calibration under distribution shift remains a key challenge and several sources of uncertainty jointly influence prediction quality. Adapting $\text{MIST}_{QR}$'s coverage under distribution shifts is an interesting and important area of future work.

## 5.3 Mitigation Strategies

Several practical solutions can alleviate these limitations. To handle larger MI values than those seen during training, one can train variants that predict normalized quantities instead of raw MI. We experimented with such variants in Appendix C.1 and observe slightly worse performance compared to standard MIST, but such variants may be preferable in scenario where large MI values can be expected. For high-dimensional data, we recommend projecting inputs to a dimension below 32 before applying MIST.

**Detecting Datasets Out of Meta Distribution**: To help assess when MIST predictions can be trusted, we describe in Appendix D a simple approach to estimate how far a test dataset lies from the meta-training distribution by leveraging MIST's internal representations. We compute distances in the learned embedding space at the last ISAB layer between the test dataset embedding and a reference set of training dataset embeddings extracted from the same layer. We find that this distance provides a reliable proxy for both prediction error and calibration degradation. Concretely, when the test dataset embedding is far from those of representative training datasets, both prediction quality and uncertainty calibration degrade predictably (Spearman rank correlation of 0.69 with MSE degradation and $-0.33$ with confidence interval coverage). This yields a practical criterion to assess when MIST predictions and confidence intervals can be trusted.

## 6    Discussion

This work rethinks the design of mutual information estimators. Rather than deriving estimators analytically from variational or density-based formulations, we treat estimation as a supervised learning problem: the estimator is a learned function, trained end-to-end on a large synthetic meta-dataset with known ground-truth MI values. This approach represents a deliberate shift in paradigm, trading theoretical motivations for a flexible and data-driven design.

**Controllability and adaptability.** A key strength of this framework lies in its controllability. Our estimators' behavior are largely determined by the synthetic datasets created; practitioners can customize, interrogate, and design datasets to improve estimator quality and reliability. Similarly, desired characteristics can be built into the learning objective, as seen in our $\text{MIST}_{\text{QR}}$ model. Since mutual information is invariant under invertible transformations, our meta-dataset can be mapped to a target modality using normalizing flows without altering its ground-truth MI. This property enables the generation of diverse synthetic meta-datasets that reflect the characteristics of arbitrary data domains, such as images, text, tabular data, or multimodal signals. By tailoring the training data through such transformations, one can design estimators specialized for particular classes of distributions or downstream applications. This mechanism opens the door to what may be viewed as a *foundational model for information-theoretic estimation*, one that can be adapted via fine-tuning to specific tasks and modalities.

**Computational efficiency.** Once trained, our learned estimators perform inference in a single forward pass (or two to obtain a confidence interval using the quantile regression model), making them several orders of magnitude faster than existing neural estimators. By amortizing the cost of inference during meta-training, our estimators achieve speedups of three to four orders of magnitude compared to neural MI estimators, and one order of magnitude compared to classical KSG implementations. The efficiency of our estimator offers new opportunities for MI estimation in settings where existing estimators are infeasible or intractable.

**Outlook.** Beyond empirical gains, the proposed framework opens numerous directions for research in both information theory and statistical learning. First, it allows the inclusion of theoretical results as inductive biases, for instance, explicitly penalizing violations of the Data Processing Inequality. Second, because MI estimators are often used within optimization loops (e.g., to maximize or minimize information), it is possible to train models that focus on *relative ordering* rather than absolute accuracy, for example, by using ranking losses that emphasize correct ordering across a batch of datasets. Taken together, these elements outline a research program: the development of trainable, meta-learned estimators as flexible, efficient, and customizable alternatives to classical estimators.

## Author Contributions

German Gritsai, Megan Richards, and Maxime Méloux provided equal contributions and share joint first authorship. German Gritsai designed the MIST architecture and learning framework, trained many of the MIST models, and developed the evaluation code used throughout the experiments. Megan Richards designed the feature selection experiments (Section 4.8), provided experiment feedback, and led the paper writing, including the documentation of existing methods (Appendix A). Maxime Méloux performed final-scale training and evaluation on the extended test set, and performed the ablated training experiments in Sections 4.5 and 4.7. Kyunghyun Cho advised the project, contributing to the project design and paper framing, as well as the idea and code for the quantile regression used in $\text{MIST}_{QR}$. Maxime Peyrard closely advised the project, providing guidance on the project design, architecture design, experimental setup, result interpretation, and paper framing.

## Acknowledgments

We thank NYU HPC for their generous support and help in computational simulations and experiments. This material is based upon work supported by the National Science Foundation Graduate Research Fellowship under Grant No. DGE-2234660 (MR). Any opinion, findings, and conclusions or recommendations expressed in this material are those of the authors(s) and do not necessarily reflect the views of the National

Science Foundation. This work was supported by the Institute of Information & Communications Technology Planning & Evaluation (IITP) with a grant funded by the Ministry of Science and ICT (MSIT) of the Republic of Korea in connection with the Global AI Frontier Lab International Collaborative Research. This work was also supported by the Samsung Advanced Institute of Technology (under the project Next Generation Deep Learning: From Pattern Recognition to AI) and the National Science Foundation (under NSF Award 1922658). This work was partially conducted within French research unit UMR 5217 and was supported by CNRS (grant ANR-22-CPJ2-0036-01) and by MIAI@Grenoble-Alpes (grant ANR-19-P3IA-0003). It was granted access to the HPC resources of IDRIS under the allocation 2025-AD011014834 made by GENCI.

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

# A Appendix

## A.1 Definitions of Mutual Information

Several equivalent expressions of mutual information are leveraged by existing estimators. For clarity, we provide a simple overview of these expressions below. For two continuous random variables $X$ and $Y$, the mutual information (MI) is defined in terms of the joint distribution $P_{(X,Y)}$ and marginal distributions $P_X$ and $P_Y$:

$$I(X;Y) = \int_{\mathcal{Y}} \int_{\mathcal{X}} P_{(X,Y)}(x,y) \log \left[ \frac{P_{(X,Y)}(x,y)}{P_X(x)P_Y(y)} \right] dx\, dy \tag{5}$$

**Donsker-Varadhan**   A commonly used expression of MI is in terms of the Kullback-Leibler (KL) divergence between the joint distribution and the product of marginals (Donsker & Varadhan, 1975):

$$I(X;Y) = D_{\mathrm{KL}}(P_{(X,Y)} \| P_X \otimes P_Y) \tag{6}$$

This expression can be understood as measuring by how much the joint distribution deviates from the product of marginal distributions. If $X$ and $Y$ are independent, $P_{(X,Y)} = P_X \otimes P_Y$, producing a KL-divergence (and MI estimate) of zero.

**Conditional Entropy**   Another common expression of MI is in terms of the conditional entropy: $H(\cdot|\cdot)$ denotes conditional entropy, and $H(X,Y)$ is the joint entropy.

$$I(X;Y) = H(X) - H(X|Y) = H(Y) - H(Y|X) = H(X) + H(Y) - H(X,Y) \tag{7}$$

This expression of MI can be understood as a measure of the reduction in uncertainty about one variable given the other.

## A.2 MI Estimators

We provide an overview of prominent MI estimators for reference in Table 2, categorized by the type of estimation they perform, the MI definition considered, and the specific estimation task. Previous works have often adopted the terminology introduced by (Song & Ermon, 2019). This framework categorizes variational estimators as either discriminative or generative: generative methods estimate the densities individually and discriminative methods estimate the ratio between joint and marginal densities.

Given the diversity of recent estimation approaches, we broaden this categorization into density and density ratio estimators. In this categorization, **density estimators** estimate MI by approximating the *form or properties of densities* $P_X, P_Y,$ and $P_{X,Y}$. This includes learning the densities explicitly with a generative model (GM, KDE, VCE), approximating densities through k-nearest-neighbor distances (KSG), learning flows to transform the marginal distributions to a tractable distribution (MIENF), or learning score functions (MINDE). In contrast, **density ratio estimators** estimate MI by modeling the *relationship between densities*, primarily in the form of the density ratio $log(\frac{P_{X,Y}}{P_X P_Y})$, without modeling the densities individually. This estimation can be performed by learning classifiers ('critics') to distinguish joint and marginal samples (MINE, SMILE, InfoNCE), or learning coupling transforms that map between the joint and marginal product (FMMI).

We chose this naming convention to better capture the diversity of existing methods and highlight the core task performed by each method. We hope that this naming convention can help avoid reader confusion between the modeling framework and the core MI estimation task. For example, works like FMMI are generative in terms of modeling choices (flows), but discriminative in terms of the MI estimation task. Similarly, many classical estimators are not generative in terms of modeling choices (binning, KSG), but are generative in terms of MI estimation. To improve clarity, we group methods by their estimation task (estimate a density, or a density ratio), regardless of the approach used to perform the estimation. Still,

this categorization, as with any abstraction, removes some nuance. We provide this appendix section as a resource for those looking to gain a better holistic understanding of existing approaches, and encourage readers to read the respective papers for precise details.

Table 2: Summary of prominent methods for estimating mutual information, along with the MI definition used and their estimation tasks.

| Method | Type | $I(X;Y)$ | Estimation Task |
|---|---|---|---|
| Binning | Density | $= \sum_{x,y} p(x,y) \log \frac{p(x,y)}{p(x)p(y)}$ | Estimate $p(x,y)$, $p(x)$, $p(y)$ via histograms |
| KSG | Density | $= H(X) + H(Y) - H(X,Y)$ | Implicitly estimate $H(\cdot)$ by writing MI's entropy definition in terms of KNN distances |
| KDE | Density | $= \sum_{x,y} p(x,y) \log \frac{p(x,y)}{p(x)p(y)}$ | Estimate $p(x,y)$, $p(x)$, $p(y)$ via kernels |
| MIENF | Density | $\approx I(f_X(X); f_Y(Y))$ | Learn flows $f_X$, $f_Y$ to Gaussian marginals |
| LMI | Density | $\approx I_{KSG}(Z_x; Z_y)$ | Learn embeddings $Z_x = g_X(X)$, $Z_y = g_Y(Y)$ via cross-predictive autoencoders $g_X, g_Y$, then apply KSG |
| GM | Density | $= \int \int p(x,y) \log \frac{p(x,y)}{p(x)p(y)} dx\, dy$ | Estimate $p(x,y)$, $p(x)$, $p(y)$ via VAEs (or flows) |
| MINDE | Density | $= D_{KL}(p(x,y)\|p(x)p(y))$ $\approx \|\nabla \log p(x,y) - \nabla \log p(x)p(y)\|^2$ | Learn scores $\nabla \log p(x,y)$, $\nabla \log p(x)$, $\nabla \log p(y)$ with a diffusion model |
| VCE | Density | $= -H[c(\mathbf{u}_X, \mathbf{u}_Y)]$ | Learn marginal ranks $\mathbf{u}_X$, $\mathbf{u}_Y$ with flows, estimate vector copula $c$ with MLE over mixture of Gaussian copulas |
| MINE | Density Ratio | $\geq \sup_\theta \mathbb{E}_{P_{X,Y}}[T_\theta] - \log(\mathbb{E}_{P_X \otimes P_Y}[e^{T_\theta}])$ | Learn critic $T_\theta(x,y)$ to maximize bound, approximating $\log(\frac{p(x,y)}{p(x)p(y)})$ |
| NWJ | Density Ratio | $\geq \sup_\theta \mathbb{E}_{P_{X,Y}}[T_\theta] - \mathbb{E}_{P_X \otimes P_Y}[e^{T_\theta - 1}]$ | Learn critic $T_\theta(x,y)$ to maximize bound, approximating $\log(\frac{p(x,y)}{p(x)p(y)}) + 1$ |
| InfoNCE | Density Ratio | $\geq \mathbb{E}_{(x,y) \sim P_{X,Y}} \left[ \log \frac{f_\theta(x,y)}{\frac{1}{n} \sum_{j=1}^n f_\theta(x,y_j)} \right]$ | Learn classifier critic $f_\theta(x,y) \propto \frac{p(x,y)}{p(x)p(y)}$ |
| SMILE | Density Ratio | $\geq \mathbb{E}_{P_{X,Y}}[T_\theta] - \log \mathbb{E}_{P_X \otimes P_Y}[\text{clip}_\tau(e^{T_\theta})]$ * | Learn classifier critic $T_\theta(x,y)$ to maximize bound, approximating $\log(\frac{p(x,y)}{p(x)p(y)})$ |
| FMMI** | Density Ratio | $= -\mathbb{E}_{T \sim \mathcal{U}[0,1]}[\text{div } v(Z_T, T)]$ $Z_0 \sim P_X \otimes P_Y,\ Z_1 \sim P_{X,Y}$ | Train velocity network $v(z,t)$ via flow matching, estimate MI as negative expected divergence of $v$ |
| InfoNet | Density Ratio | $\geq \sup_\theta \mathbb{E}_{P_{X,Y}}[\theta] - \log(\mathbb{E}_{P_X \otimes P_Y}[e^\theta])$ | Pretrain network $F(x,y) \to \theta$, where $\theta$ is MINE's optimal discriminant as as a look-up table |
| MIST | Meta-Learned | $\approx F(x,y)$ | Learn estimator $F$ by direct optimization |

\* $\text{clip}_\tau(z) \in [e^{-\tau}, e^\tau]$.

\*\* FMMI learns a coupling transform between joint and marginal distributions. This estimation implicitly encodes the density ratio, which could be derived from the velocity field with an integration. However, FMMI does not perform this integral and does not estimate the density ratio at each point. Instead, FMMI estimates MI as an expectation of the field's divergence. FMMI authors describe their method as a hybrid method. We categorize FMMI as a density ratio method due to the focus on learning the relationship between joint and marginal distributions, rather than modeling the densities individually.

### A.3 Empirical Evaluation Scope of MI Estimators

In Table 3, we provide a summary of how prominent estimators were empirically evaluated. We only include the evaluations of estimators in the papers introducing the method, as we are most interested in how estimator quality is determined. This summary focuses on the experiments evaluating the estimator directly, and excludes evaluations in downstream settings, where ground truth MI values are not available.

Table 3: Summary of the empirical evaluation scope of prominent mutual information estimators. Comma-separated entries indicate values for different experimental settings (e.g., KSG has four experiments). Dashes indicate data that is not available or not specified.

| Method | Data Type | $\# \mathcal{D}$ | $\# \not\sim \mathcal{N}$ | $d_{\max}$ | $I_{\max}$ | $n_{\min}$ |
|---|---|---|---|---|---|---|
| Binning | Time-Series | 1 | 1 | 1 | - | 65K |
| KSG | Synthetic | 1,3,1,1 | 0,3,0,0 | 2,1,3,8 | 0.83,1,1,1.2 | 125,1K,10K,50K |
| KDE | Time-Series | 4 | 4 | 1 | - | 400 |
| MIENF | Synthetic | 1,5 | 0,5 | 1024,32 | 10,10 | 10K,10K |
| LMI | Synthetic,Images,PLM | 1,1,1,1 | 0,0,1,1 | 5K,50,784,1024 | 2,1,1,1 | 2K,100,5K,4.4K |
| GM | Synthetic | 2 | 1 | 20 | 10 | 256K |
| MINDE | Synthetic | 19,3 | 16,2 | 25,9 | 1,5 | 100K,- |
| VCE | Synthetic,Images,Text | 5,5,2,2 | 4,3,1,2 | 64,64,16,16 | 16,14,7,2.1 | 10K,10K,10K,4K |
| MINE | Synthetic | 2,3 | 0,3 | 20,2 | 40,- | -,- |
| NWJ | - | - | - | - | - | - |
| InfoNCE | - | - | - | - | - | - |
| SMILE | Synthetic | 2 | 1 | 20 | 10 | 256K |
| FMMI | Synthetic | 4 | 2 | 160 | 115 | 10K |
| InfoNet | Synthetic | 1,20,10,3 | 0,19,9,0 | 1,1,1,128[1] | 0.9,0.9,-,- | 2K,2K,2K,50 |
| MIST | Synthetic | 23 | 19 | 32 | 40 | 10 |

[1] Using slicing, which takes an expectation over random 1D projections

Early methods for mutual information estimation, such as **binning**, were evaluated primarily using simulated time-series data. These commonly included sine and cosine functions, as well as chaotic dynamical systems such as the Lorenz and Rossler attractors (where the task is phase reconstruction, and no ground truth is available). In the initial histogram binning work (Fraser & Swinney, 1986), the Rossler attractor was used. For this work, we include it in the table as having one 1D distribution, which is non-Gaussian, with no ground truth MI reported (and therefore no maximum). The experiment calculated the mutual information over 65,536 points. **KDE** (Moon et al., 1995) was similarly evaluated with time-series data. KDE was evaluated on 1D synthetic sine waves, an autoregressive function of a sine and a Gaussian distribution, the Lorenz attractor, and the Rossler attractor. 400 samples were used for the sine function, and 500 samples were used for the autoregressive function. 4096 and 2048 samples were used for the Lorenz and Rossler attractors, respectively. Estimators are evaluated on the quality of the reconstruction, with no ground truth MI available. **KSG** (Kraskov et al., 2004) performed evaluation on 2D correlated Gaussians with unit variance and zero mean, testing sample sizes of 125, 250, 500, 1K, 2K, 4K, 10K, and 20K. The MI was 0.830366. Authors performed error analysis with 10K samples, and performed experiments on the gamma-exponential distribution, the ordered Weinman exponential, and the circle distribution. All of these had MI values below 1, and considered samples with 1K and 10K data points. For their higher dimension experiments, the estimator was evaluated on 3-D Gaussian distributions with a sample size of 10K in one experiment, and on m-correlated uniform Gaussians up to dimension 8 with a sample size of 50K. We include KSG in the table as being evaluated on synthetic 4 distributions (Gaussian, Weinman-exponential, gamma-exponential, and circle), 3 of which are non-Gaussian. The maximum dimension considered was 8, while the maximum MI values were 0.83, 1, and 1.2 for the 1-D Gaussians, 3-D Gaussians, and 8-D Gaussian experiments, respectively. The minimum sample size in these experiments was 125 for the 1-D Gaussian, 10K for other distributions, and 50K for larger dimensions. **MIENF** (Butakov et al., 2024) first evaluated their estimator on 2D correlated Gaussian distributions mapped into higher-dimensional images (16x16 and 32x32), evaluated on 10K samples, and with MI values between 0 and 10. In their second experiment, authors

evaluated their method on five non-compressible distributions (uniform, smoothed uniform, student (dof = 4), arcsignh (student) student with dof = 1,2). For this experiment, authors varied the dimensions of these distribution from 2 to 32, using 10K samples for each prediction. The MI was set to 10 for all distributions and dimensions.

**Latent-MI** (LMI; Gowri et al., 2024) was first evaluated on multivariate Gaussian distributions under varying original data dimensionality and latent dimensionality (the dimensionality of the KSG input). Their multivariate Gaussian experiment considered up to 10 latent dimensions and 5K original dimensions, with 2K samples. The MI ranged from 0 to 2 bits. In a second experiment, the authors studied the sample requirements of their estimators. For this experiment, authors used multivariate Gaussians from dimension 1 to 50, each with one correlated dimension, and an MI of 1. Authors varied the sample size from 100 to 10K. In a third experiment, MNIST and protein embeddings were resampled to align with a Bernoulli vector with set correlation. We consider the MIST and protein embedding experiments separately. For MNIST, the dimension was 784, and the sample size was 5K, and the dataset was generated with MI values between 0 and 1. For the protein embeddings, the dimension was 1024, the sample size was 4402, and the dataset was generated with MI values between 0 and 1.

For **GM** and **SMILE** (Song & Ermon, 2019), evaluation was initially performed two distributions. The first distribution was a 20-D Gaussian with correlation $p$ (borrowing from a setup in (Belghazi et al., 2018)), and the second was this distribution with a cubic operation applied. Authors follow a setup from (Poole et al., 2019), in which MI values were increased from 0 to 10 at step intervals during the critic training process. Training occurred over 20K steps, with MI increasing by 2 every 4K steps. The batch size was 64, and each batch was a new random sample. For the sake of our table, we consider the number of samples used for estimation to be the number of samples used to fit the critic for a given MI value. Since the estimator saw 4K steps of batch size 64 for each MI (2, 4, 6, 8, 10), we list the minimum samples seen as 4,000 x 64 = 256K. In a second set of experiments, both methods are evaluated for consistency tests with pairs of MNIST and CIFAR images. Because these do not have ground truth MI values, we exclude them from the table. In this work, a flow-based GM model is used for the first experiment, and a VAE-based GM model is used for the consistency experiment.

**MINDE** (Franzese et al., 2023) used the BMI benchmark (Czyż et al., 2023) to evaluate their model. BMI contains 40 tasks. We count any unique combination of distribution and transform (including degrees of freedom and noise levels) to be a unique distribution type, excluding tasks that only differ by the dimensions or by correlation strength. Among the 40 tasks, there are 19 of these unique distributions, and 16 of these are non-normal. In their first experiment, they evaluate on these distributions with 100K samples, with MI ranging from 0.2 to 1. In a second experiment, they evaluate on 3x3 multivariate normal distributions, along with a half-cube and spiral transformation of them. Their experiment increases MI from 0 to 5. We count this experiment as having 3 unique distribution types, 2 of which are non-normal. The sample size is not directly reported. In a third experiment, they repeat the consistency tests from SMILE (Song & Ermon, 2019) on MNIST only, which we exclude because there is no ground truth MI.

**VCE** Chen et al. (2025) first evaluate estimators on five synthetic distributions, under varying dependence. In this setting, MI ranges from 0 to 16, 10K samples are used, and dimensions range from 2 to 64. In a second experiment, the authors study increasing dimensionality on five distributions (three non-normal), with dimensions from 2 to 64 and MI values up to 14. 10K samples were used. In a third experiment, authors consider correlated rectangles and gaussian plates. These images have dimension 16x16 but are processed by an autoencoder prior to prediction to produce 16 dimensional embeddings. The embedding portion is considered a pre-processing step, rather than part of the estimation method (as in LMI), and we therefore report the embedding dimension. This is consistent with other similar experiments, in which lower-dimensional data is scaled up to higher dimensions prior to evaluation, for which we correspondingly select the scaled dimensionality. The dimension reported is the dimension of data provided to the estimator. We consider this experiment to have 2 distributions, 1 of which is non normal, and dimension of 16. The maximum MI value is 7, and 10K samples used. In a fourth experiment, they consider language embeddings from Llama-3 13B and Bert. We consider this to be 2 non-normal distributions. Data is similarly processed by an autoencoder to be 16 dimensions. The underlying MI is estimated with resampling, and the maximum reported value is 2.1. 4K samples were used.

**MINE** (Belghazi et al., 2018) was first evaluated on 2-D and 20-D Gaussians with MI from 0 to 2 and 0 to 40, respectively. The number of samples used was not specified. In a second experiment, the authors applied 3 deterministic nonlinear transformations to a normal distribution and demonstrated that their estimates are invariant to it. MINE then performs several downstream application experiments. **NWJ** (Nguyen et al., 2010) introduced a variational bound for estimating the KL-divergence between two arbitrary distributions, and their experiments focused on evaluating the quality of this bound and estimation on various distributions. The authors did not evaluate this approach on mutual information prediction directly, for which the distributions compared would be a joint distribution and the product of its marginals. **InfoNCE** (Oord et al., 2018) was developed as a method for representation learning, with evaluation including representation learning tasks but without any evaluation of MI prediction itself.

**FMMI** (Butakov et al., 2025) empirically evaluate their method on four distribution types (correlated normal, half-cube correlated normal, correlated uniform, smoothed uniform), with MI ranging from 0 to 80 nats. We convert nats (natural logarithm, base $e$) to bits (base 2), by dividing the nats value by $\log_e(2)$, and round to the nearest whole number (115). In this experiment, the training size was 100K samples, and the test set was 1K samples. The dimensions ranged from 0 to 160 dimensions. FMMI is also applied to predict O-information in fMRI data.

**InfoNet** (Hu et al., 2024) perform their first experiment evaluating test-time efficiency on GMM distributions of various lengths, using sequence of 200, 500, 1000, 2000, and 5000 samples. Since no additional details about the estimation are included (maximum MI, dimension, number of samples), we exclude it from the table. In their second experiment (the first we report in the table), they benchmark MI estimation on 1D gaussian distributions with varying correlation. The MI ranges from 0 to 0.9, and 2000 samples are used. In the third experiment (the second we report in the table), the authors evaluate their method on 1D GMMs with multiple components (up to 20, which we consider as 20 distinct distribution types, 19 of which are non-gaussian), using 10 MI levels from 0 to 0.9. Authors use 2K samples. In their fourth experiment (the third we report in the table), the authors evaluate order accuracy on 1D GMMs with components from 1 to 10 (we count this as 10 distinct distributions, 9 of which are non-normal). The number of samples and maximum MI value are not reported. In their fifth experiment (the fourth we report in the table), they evaluate an application in high-dimensional independence testing, using slicing to scale their 1D-trained model. Their experiment tests three structures of correlation in gaussian distributions, and we report this as 3 distribution types, 0 of which are non-gaussian. They evaluate on 16 and 128 dimensions, and the number of samples varied from 50 to 950. In their sixth experiment, they evaluate their approach using MI to perform object segmentation in video data. Because this experiment is a downstream application without ground-truth MI values, we exclude it from the table.

In this work, **MIST**, we use the BMI library to generate a range of 23 distribution types (16 in $\mathcal{M}_{\text{train}}$, 7 in $\mathcal{M}_{\text{test,OoMD}}$). Of these distributions, 19 are non-normal. The maximum dimension we use is 32, and we test MI values up to 40. The minimum sample size in our experiments is 10. Our downstream experiments in consistency tests and feature selection are excluded from this table, whose focus is evaluating MI estimation directly.

## B Details of Meta-Dataset Creation

We construct a meta-dataset of synthetic distributions with known MI using the BMI library (Czyż et al., 2023), which generates complex yet tractable distributions via invertible transformations applied to base families with analytical MI. Each meta-sample consists of joint samples from a distribution paired with its ground-truth MI value, enabling supervised learning directly over distributions.

To promote diversity and generalization, we partition distribution families into disjoint training $\mathcal{D}_{\text{train}}$ and testing $\mathcal{D}_{\text{test}}$ groups and vary the sample dimensionality from 2 to 32. Since many existing estimators are prohibitively slow at this scale, we construct two evaluation corpora: a smaller $\mathcal{M}_{\text{test}}$ benchmark set for comparison with existing methods, and a larger $\mathcal{M}_{\text{test-extended}}$ set for assessing generalization to novel and higher-dimensional distributions. Complete details of meta-dataset sizes and included distribution families are provided in Table 4.

The dataset includes both *In-Meta-Distribution* (IMD) and *Out-of-Meta-Distribution* (OoMD) families, differing in their base distributions and MI-preserving transformations.

**Distribution families**

The meta-dataset comprises samples drawn from the following base distributions, each with its own structured variants and associated sampled hyperparameters:

- **multi_normal**: Multivariate normal distributions over $(X, Y)$ with jointly Gaussian structure and identity marginal variances. Structured variants are parameterized as follows:
  - `dense`: `off_diag` $\sim \mathcal{U}(0.0, 0.5)$     (uniform pairwise correlation between all variables)
  - `lvm`: `n_interacting` $\in \{1, \ldots, 5\}$     (latent dimensionality)
    `alpha` $\sim \mathcal{U}(0.0, 1.0)$     (anisotropy; relative scaling of $X$ vs. $Y$ loadings)
    `lambd` $\sim \mathcal{U}(0.1, 3.0)$     (latent signal strength)
    `beta` $\sim \mathcal{U}(0.0, 1.5)$     (additive noise scale in $X$)
    `eta` $\sim \mathcal{U}(0.1, 5.0)$     (intensity of nonlinear warping in $X$)
  - `sparse`: `n_interacting` $\in \{1, \ldots, min(5, d)\}$   (number of correlated pairs, governs low-rank structure)
    `strength` $\sim \mathcal{U}(0.1, 5.0)$     (latent signal strength, used as `lambd` and `eta`)
    The covariance is constructed via a `GaussianLVMParametrization` with $\alpha = \beta = 0$, $\lambda = \eta =$ `strength`, so that only `n_interacting` pairs $(X_i, Y_i)$ are correlated.
  - `2pair`: A special case of `sparse` with `n_interacting` $= 2$ fixed.

- **multi_student**: Multivariate Student's $t$-distributions over $(X, Y)$ sharing the same covariance (or scatter) structure as their Gaussian counterparts, but with heavier tails controlled by the degrees of freedom. Structured variants are parameterized as follows:
  - `dense`: `off_diag` $\sim \mathcal{U}(0.0, 0.5)$     (uniform pairwise correlation)
    `df` $\sim \mathcal{U}(1, 10)$     (degrees of freedom; lower values correspond to heavier tails)
  - `sparse`: `n_interacting` $\in \{1, \ldots, min(5, d)\}$     (number of correlated pairs)
    `strength` $\sim \mathcal{U}(0.1, 5.0)$     (latent signal strength, used as `lambd` and `eta`)
    `df` $\sim \mathcal{U}(1, 10)$     (degrees of freedom)
  - `2pair`: A special case of `sparse` with `n_interacting` $= 2$ fixed.

- **multi_additive_noise**: A multivariate additive noise model in which, for each dimension $j = 1, \ldots, d$,
$$X_j \sim \text{Uniform}(0, 1), \quad N_j \sim \text{Uniform}(-\epsilon, \epsilon), \quad Y_j = X_j + N_j,$$
  with a shared noise scale $\epsilon \sim \mathcal{U}(0.1, 2.0)$. This distribution is non-Gaussian with bounded support, and its mutual information equals the sum of per-dimension contributions.

  Sampled hyperparameter:
  - `epsilon` $\sim \mathcal{U}(0.1, 2.0)$     (half-width of the uniform noise interval, controls the signal-to-noise ratio)

The true MI values for these distributions and their transformations are computed analytically or numerically, depending on tractability.

**MI-Preserving Transformations**

Each base distribution is composed with a MI-preserving product diffeomorphism $f \times g$ applied independently to the $X$ and $Y$ marginals. The following transformations are used:

- **base**: The identity mapping (no transformation).

- **asinh**: The inverse hyperbolic sine, applied element-wise, which compresses the tails of heavy-tailed distributions.

- **halfcube**: A signed power-$\frac{3}{2}$ mapping, applied element-wise, which expands the tails relative to the center, introducing non-Gaussian marginal shapes.

- **wigglify**: A quasi-periodic perturbation of the identity, applied element-wise with different coefficients for $X$ and $Y$. These invertible maps introduce high-frequency, non-linear structure while preserving the global ordering of values.

**Meta-Dataset Composition**

We report in Table 4 the exact composition of the training and test splits of the meta-datasets in terms of distribution families, parameterizations, and MI-preserving transforms.

Table 4: Meta-Dataset Description. Each distribution family is labeled as `multi_<base_dist>-<structure>-<transform>`, where `<base_dist>` denotes the base distribution type, `<structure>` specifies the parameterization structure, and `<transform>` indicates the nonlinear transformation applied to the data. For resource-constrained evaluation, a reduced test subset of 1k samples is used for both IMD and OoMD splits. The reduced subsets preserve the same distribution family composition as their full counterparts.

| Dataset Split | Size (Extended / General) | Distribution Families |
|---|---|---|
| $\mathcal{M}_{\mathbf{train}}$ | 624652 | `multi_normal-dense-[base, asinh, halfcube]`, `multi_normal-lvm-[base, wigglify]`, `multi_normal-sparse-[base, wigglify]`, `multi_normal-2pair-base`, `multi_student-dense-[base, wigglify]`, `multi_student-sparse-[base, asinh, halfcube]`, `multi_student-2pair-[asinh, halfcube]`, `multi_additive_noise-wigglify` |
| $\mathcal{M}_{\mathbf{test,\ IMD}}$ | 372000 / 1080 | `multi_normal-dense-[base, halfcube]`, `multi_normal-lvm-wigglify`, `multi_student-dense-base`, `multi_student-sparse-[asinh, base]` |
| $\mathcal{M}_{\mathbf{test,\ OoMD}}$ | 434000 / 1260 | `multi_normal-dense-wigglify`, `multi_normal-sparse-[asinh, halfcube]`, `multi_student-dense-halfcube`, `multi_student-sparse-wigglify`, `multi_additive_noise-[base, halfcube]` |

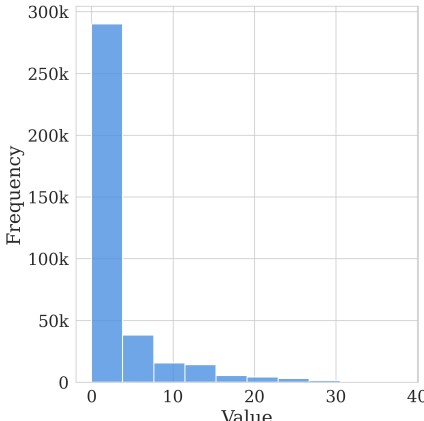

Figure 10: Distribution of true MI values within $\mathcal{M}_{\text{train}}$.

Table 5: Mean MI estimates across prediction strategies on $\mathcal{M}_{\text{test}}$ (dim = 9), comparing normalization methods and coefficient configurations. "3-coef" uses the estimators $(\hat{H}_{X,\text{inv}}, \hat{H}_{Y,\text{inv}}, \hat{H}_{XY,\text{inv}})$; the last two entries show predictions with $\hat{I}_{\text{inv}}$, with and without the physics-informed loss $\mathcal{L}_{\text{phys}}$. Values are (mean $\pm$ stddev) over three runs.

| Prediction Method | $\mathcal{M}_{\text{test,IMD}}$ | $\mathcal{M}_{\text{test,OoMD}}$ |
|---|---|---|
| Direct MI | $\mathbf{5.4e^{-2}}$ ±2.0 | $\mathbf{1.2e^{0}}$ ±2.3 |
| Log norm. $(\hat{I}_{\log})$ | $8.5e^{0}$ ±2.1 | $1.4e^{1}$ ±2.4 |
| Inverse norm. $(\hat{I}_{\text{inv}})$ | $8.7e^{-2}$ ±2.5 | $1.9e^{0}$ ±3.3 |
| Entropy (3-coef) | $2.5e^{-1}$ ±3.0 | $3.5e^{0}$ ±4.0 |
| Entropy & $\hat{I}_{\text{inv}}$ (with $\mathcal{L}_{\text{phys}}$) | $\mathbf{5.2e^{-2}}$ ±1.8 | $\mathbf{1.5e^{0}}$ ±2.0 |
| Entropy & $\hat{I}_{\text{inv}}$ (w/o $\mathcal{L}_{\text{phys}}$) | $7.0e^{-2}$ ±2.4 | $1.8e^{0}$ ±3.0 |

## C  Results

### C.1  Experiments for Normalized and Multi-Task Predictions

In high-dimensional settings, estimated MI values increased substantially, often reaching several dozen, as illustrated in Figure 10. Such a wide numerical range can make direct regression with MSE or QCQR losses unstable, since large target values can lead to unstable gradients and slow convergence. To mitigate this issue, we applied normalization to the target variable, constraining its range and improving numerical stability.

To stabilize regression over wide MI ranges, we evaluated two normalization schemes:

$$\hat{I}_{\log} = \log\big(I(X;Y) + 1\big), \qquad\qquad \hat{I}_{\text{inv}} = \frac{I(X;Y)}{I(X;Y) + 1}, \qquad (8)$$

which reduce the impact of outliers and transform regression into a constrained prediction problem.

In addition to target normalization, we enriched the model by predicting auxiliary entropy coefficients corresponding to $H(X)$, $H(Y)$, and $H(X,Y)$. Using the same monotonic mapping $f(z) = z/(1 + z)$, the normalized forms were defined as:

$$\hat{H}_{X,\text{inv}} = f\big(H(X)\big), \qquad \hat{H}_{Y,\text{inv}} = f\big(H(Y)\big), \qquad \hat{H}_{XY,\text{inv}} = f\big(H(X,Y)\big). \qquad (9)$$

We investigated two formulations: ($i$) predicting the three entropy coefficients and reconstructing MI via Eq. 7, and ($ii$) predicting all four coefficients $\big(\hat{I}_{\text{inv}}, \hat{H}_{X,\text{inv}}, \hat{H}_{Y,\text{inv}}, \hat{H}_{XY,\text{inv}}\big)$ in the multi-task setting.

For the four-coefficient configuration, a physics-informed term was introduced to enforce consistency between two independent MI estimates. The first estimate is obtained from the directly predicted MI coefficient $\hat{I}_1 = f^{-1}\big(\hat{I}_{\text{inv}}\big)$ and the entropy-derived target $\hat{I}_2 = f^{-1}\big(\hat{H}_{X,\text{inv}}\big) + f^{-1}\big(\hat{H}_{Y,\text{inv}}\big) - f^{-1}\big(\hat{H}_{XY,\text{inv}}\big)$. To achieve alignment between these quantities, the overall training objective is expressed as

$$\mathcal{L}_{\text{total}} = \underbrace{\frac{1}{4} \sum_{j \in \{\text{inv}, X, Y, XY\}} \big\|\hat{Q}_j - y_j\big\|^2}_{\mathcal{L}_{\text{base}}} + \lambda_{\text{phys}} \underbrace{\mathbb{E}_{\mathcal{B}}\left[\left|\frac{\hat{I}_1}{\hat{I}_2 + \epsilon} - 1\right|\right]}_{\mathcal{L}_{\text{phys}}}, \qquad (10)$$

where $\hat{Q}_j \in \{\hat{I}_{\text{inv}}, \hat{H}_{X,\text{inv}}, \hat{H}_{Y,\text{inv}}, \hat{H}_{XY,\text{inv}}\}$ denotes the predicted normalized coefficients, $\lambda_{\text{phys}} = 0.3$ balances regularization strength, $\epsilon > 0$ prevents division by zero, and $\mathbb{E}_{\mathcal{B}}[\cdot]$ denotes expectation over a training batch. We trained this model on a fixed-dimension subset from $\mathcal{M}_{\text{train}}$ with all samples having dim = 9. The goal

of this setup is to study whether predicting physically-related coefficients can help refine the model's latent representations, incorporate additional structural knowledge, and improve prediction robustness.

Experiments on $\mathcal{M}_{\text{test}}$ with dim $= 9$ are summarized in Table 5. The best results were obtained using two methods: direct MI prediction, and the four-coefficient formulation with the physics-informed loss. The comparatively lower performance of the three-coefficient model can be attributed to error propagation, since each entropy term is predicted independently and their individual errors accumulate during MI reconstruction. In this work, we proceed with the direct MI prediction strategy for subsequent experiments. Nevertheless, the four-coefficient formulation remains a promising direction for future research, as it provides richer structural information about the underlying distributions and could facilitate the derivation of both MI and entropy values within a unified framework.

## C.2 Median results

We report in Table 6 additional results for the evaluation of the baseline estimators for different sample sizes (main results in Table 1), namely reporting the median (rather than mean) MSE.

Table 6: Median MSE loss values of each estimator, grouped by the number of samples $n$ given to the estimator and distributions. We **bold** (resp. underline) the best (resp. second best) row of each column.

| $n \in$ | In-Meta-Distribution (IMD) | | | Out-of-Meta-Distribution (OoMD) | | |
| | $[10, 100]$ | $[100, 300]$ | $[300, 500]$ | $[10, 100]$ | $[100, 300]$ | $[300, 500]$ |
|---|---|---|---|---|---|---|
| CCA | $5.7e^2$ | $1.7e^0$ | $6.6e^{-1}$ | $5.7e^2$ | $9.8e^{-1}$ | $1.4e^0$ |
| KSG | $\mathbf{2.0e^{-1}}$ | $6.0e^{-2}$ | $4.9e^{-2}$ | $6.2e^0$ | $7.7e^{-1}$ | $2.8e^0$ |
| MINE | $1.5e^2$ | $1.4e^0$ | $1.4e^0$ | $2.4e^2$ | $6.1e^1$ | $2.1e^1$ |
| InfoNCE | $5.9e^1$ | $9.3e^{-1}$ | $6.1e^{-1}$ | $8.8e^1$ | $2.9e^0$ | $5.1e^0$ |
| InfoNet | $3.5e^{-1}$ | $3.9e^{-1}$ | $7.7e^{-1}$ | $7.2e^0$ | $2.3e^0$ | $6.2e^0$ |
| FMMI | $2.8e^1$ | $2.5e^1$ | $1.7e^1$ | $3.3e^1$ | $2.8e^1$ | $2.2e^1$ |
| VCE | $3.7e^0$ | $2.4e^0$ | $1.5e^0$ | $3.2e^0$ | $3.0e^0$ | $2.8e^0$ |
| DV | $8.1e^8$ | $2.0e^0$ | $7.5e^{-1}$ | $1.2e^9$ | $1.9e^1$ | $1.4e^1$ |
| LMI | $\underline{2.6e^{-1}}$ | $1.7e^{-1}$ | $2.6e^{-1}$ | $7.0e^0$ | $3.7e^{-1}$ | $1.6e^0$ |
| MINDE | $9.5e^{-1}$ | $2.4e^{-1}$ | $2.1e^0$ | $9.0e^0$ | $1.2e^0$ | $3.5e^0$ |
| MIST | $3.3e^{-1}$ | $\underline{1.9e^{-2}}$ | $\underline{1.3e^{-2}}$ | $\underline{1.3e^0}$ | $\underline{1.8e^{-1}}$ | $\underline{1.6e^{-1}}$ |
| MIST$_{QR}$ | $\underline{2.6e^{-1}}$ | $\mathbf{9.1e^{-3}}$ | $\mathbf{6.9e^{-3}}$ | $\mathbf{1.0e^0}$ | $\mathbf{1.4e^{-1}}$ | $\mathbf{1.2e^{-1}}$ |

## C.3 InfoNet results

InfoNet was trained on Gaussian mixtures, which constitute less than half of $\mathcal{M}_{\text{test}}$. For a fairer comparison in-meta-distribution, we first report in Table 7 the average and median MSE of InfoNet and our estimators computed solely on the base multivariate Gaussian subset of $\mathcal{M}_{\text{train, IMD}}$ (`multi_normal-dense-base`). Additionally, we report in Table 8 the average MSE of the estimators computed on the larger subset of $\mathcal{M}_{\text{test}}$ that includes all variants and transforms of multivariate Gaussian distributions (`multi_normal-*`). This includes both IMD and OoMD distribution settings for MIST.

Table 7: Average, median and standard deviation of the MSE of InfoNet and MIST for base multivariate Gaussian distributions in $\mathcal{M}_{\text{test}}$.

| $n \in$ | Average MSE $\pm$ stddev (median) | | |
| | $[10, 100]$ | $[100, 300]$ | $[300, 500]$ |
|---|---|---|---|
| InfoNet | $\mathbf{1.3e^{-1}}$ $\pm 1.2e^{-1}$ $(7.1e^{-2})$ | $1.4e^{-1}$ $\pm 2.0e^{-1}$ $(5.0e^{-2})$ | $2.8e^{-1}$ $\pm 2.8e^{-1}$ $(1.9e^{-1})$ |
| MIST | $3.0e^{-1}$ $\pm 6.9e^{-1}$ $(2.1e^{-2})$ | $8.1e^{-3}$ $\pm 9.3e^{-3}$ $(4.9e^{-3})$ | $3.8e^{-3}$ $\pm 2.6e^{-3}$ $(3.3e^{-3})$ |
| MIST$_{QR}$ | $1.6e^{-1}$ $\pm 3.2e^{-1}$ $(\mathbf{1.4e^{-2}})$ | $\mathbf{4.3e^{-3}}$ $\pm 6.9e^{-3}$ $(\mathbf{2.5e^{-3}})$ | $\mathbf{1.5e^{-3}}$ $\pm 1.2e^{-3}$ $(\mathbf{1.2e^{-3}})$ |

Table 8: Average MSE ($\pm$ stddev) of InfoNet and MIST for all multivariate Gaussian distributions in $\mathcal{M}_{\text{test}}$.

| $n \in$ | $\mathcal{M}_{\text{test, IMD}}$ (for MIST) | | | $\mathcal{M}_{\text{test, OoMD}}$ (for MIST). | | |
| | $[10, 100]$ | $[100, 300]$ | $[300, 500]$ | $[10, 100]$ | $[100, 300]$ | $[300, 500]$ |
|---|---|---|---|---|---|---|
| InfoNet | $4.4\mathrm{e}^0$ $\pm 2.1\mathrm{e}^1$ | $8.5\mathrm{e}^0$ $\pm 3.8\mathrm{e}^1$ | $5.0\mathrm{e}^0$ $\pm 1.9\mathrm{e}^1$ | $4.8\mathrm{e}^1$ $\pm 1.2\mathrm{e}^2$ | $3.7\mathrm{e}^1$ $\pm 1.2\mathrm{e}^2$ | $7.3\mathrm{e}^1$ $\pm 1.7\mathrm{e}^2$ |
| MIST | $2.2\mathrm{e}^0$ $\pm 1.3\mathrm{e}^1$ | $1.3\mathrm{e}^{-1}$ $\pm 3.5\mathrm{e}^{-1}$ | $8.7\mathrm{e}^{-2}$ $\pm 2.8\mathrm{e}^{-1}$ | $2.9\mathrm{e}^0$ $\pm 5.2\mathrm{e}^0$ | $4.1\mathrm{e}^{-1}$ $\pm 1.3\mathrm{e}^0$ | $4.6\mathrm{e}^{-1}$ $\pm 1.1\mathrm{e}^0$ |
| MIST$_{QR}$ | $\mathbf{2.2e^0}$ $\pm 1.5\mathrm{e}^1$ | $\mathbf{1.1e^{-1}}$ $\pm 3.8\mathrm{e}^{-1}$ | $\mathbf{6.2e^{-2}}$ $\pm 2.0\mathrm{e}^{-1}$ | $\mathbf{2.6e^0}$ $\pm 5.3\mathrm{e}^0$ | $\mathbf{2.7e^{-1}}$ $\pm 5.7\mathrm{e}^{-1}$ | $\mathbf{2.1e^{-1}}$ $\pm 4.2\mathrm{e}^{-1}$ |

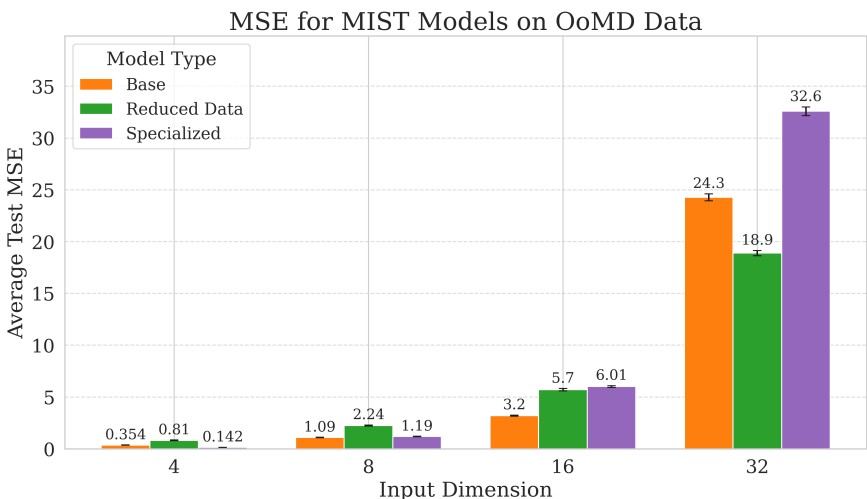

Figure 11: Average MSE obtained by variable- and fixed-dimensionality versions of MIST on subsets of $\mathcal{M}_{\text{test-extended,OoMD}}$ with fixed dimensions.

## C.4 Dimensionality Ablation

Figures 11 and 12 contain additional bar plots for the experiment described in the second half of Section 4.7. We give in Figures 13 through 15 a high-level summary of the general performance of all various fixed- and variable-dimensionality models for ease of comparison.

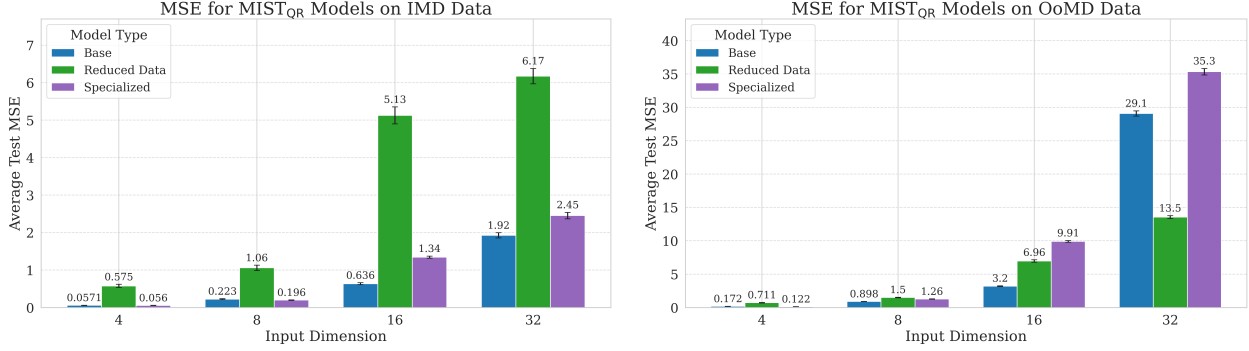

Figure 12: Average MSE obtained by variable- and fixed-dimensionality versions of MIST$_{QR}$ on subsets of $\mathcal{M}_{\text{test-extended}}$ with fixed dimensions. Left: $\mathcal{M}_{\text{test-extended,IMD}}$. Right: $\mathcal{M}_{\text{test-extended,OoMD}}$.

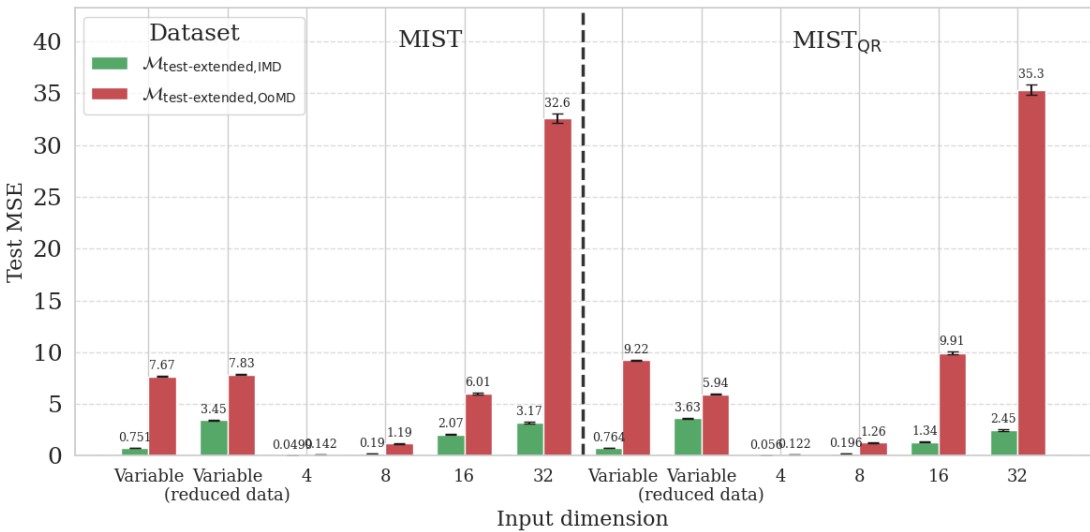

Figure 13: Average MSE of the proposed models with variable input dimensionality versus dimension-specific variants. In this plot and the following ones, the test set was similar to the training one: variable-dimensionality models were evaluated on the full $\mathcal{M}_{\text{test-extended}}$, while other models were evaluated on subsets of it.

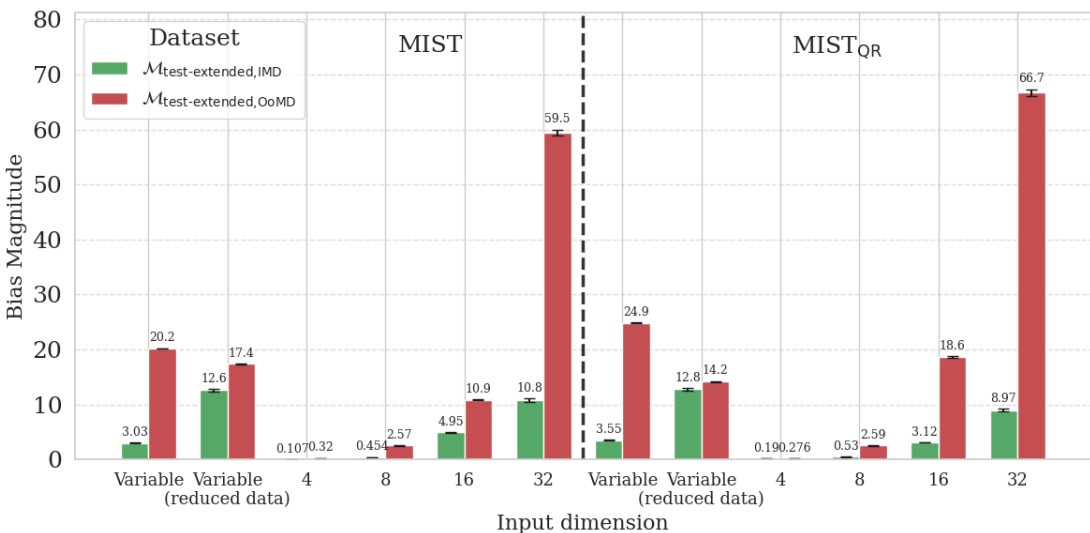

Figure 14: Average bias magnitude of the proposed models with variable input dimensionality versus dimension-specific variants.

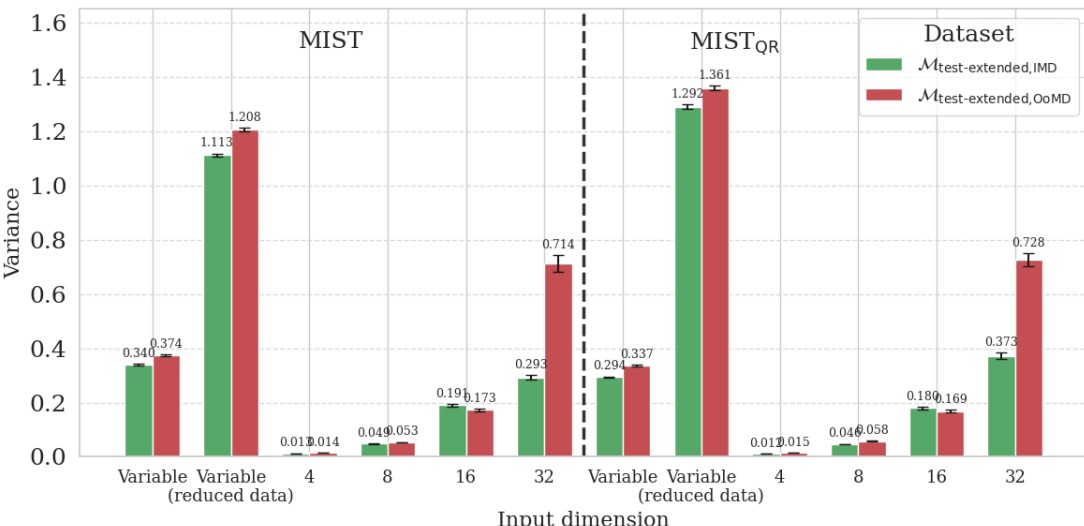

Figure 15: Average variance of the proposed models with variable input dimensionality versus dimension-specific variants.

### C.5   Comparing With Existing Estimators

For a more detailed evaluation of existing baselines and our trained models MIST and $\text{MIST}_{\text{QR}}$ on $\mathcal{M}_{\text{test}}$, we also report the MSE loss values averaged separately over each distribution family in Table 9.

### C.6   Inference Time

We report in Table 10 the inference time of each of the baselines, averaged across the data from Figure 7.

### C.7   Scaling Behavior

We report in Table 11 the performance of the main model (5.5M) used in our experiments compared to scaled-down models. We give in Figure 16 a fine-grained heatmap of the MSE performance obtained by MIST and $\text{MIST}_{QR}$ for various sample size groups and dimensions.

## D   Estimating OOMD Performance Degradation via MIST Representations

As MIST models are trained on meta-distributions of datasets, both their predictive accuracy and uncertainty estimation may degrade when evaluated on datasets that lie outside the meta-training distribution. While the main paper demonstrates good generalization performance, we further investigate a simple, non-learning-based mechanism to estimate how far a given input dataset is from the training distribution, and consequently how much one can trust the predictions and confidence intervals produced by MIST.

Our approach leverages the internal representations computed by MIST. Specifically, we use the fixed-size vector representations produced at the last ISAB layer, which serve as dataset-level embeddings. Intuitively, datasets that are considered similar by MIST are expected to have similar embeddings in this representation space.

The procedure is as follows:

- We first sample $K$ datasets from the meta-training distribution and compute their embeddings using a forward pass through a trained MIST (or MIST_$QR$) model. These embeddings form a reference set representing the training meta-dataset.

Table 9: Mean mutual information (MI) estimates across data distributions. Color indicates whether distributions are from the in-meta-distribution (blue) or out-of-meta-distribution (red) subsets from $\mathcal{M}_{\text{test}}$. We also highlight in green values that are within one order of magnitude of the best and mark the best values in **bold**.

| | Multinormal (dense) | Wiggly @ Multinormal (lvm) | Half-cube @ Multinormal (dense) | Student-t (dense) | Student-t (sparse) | Asinh @ Student-t (sparse) | Wiggly @ Multinormal (dense) | Half-cube @ Multinormal (sparse) | Asinh @ Multinormal (sparse) | Wiggly @ Student-t (sparse) | Half-cube @ Student-t (dense) | Additive Noise | Half-cube @ Additive Noise |
|---|---|---|---|---|---|---|---|---|---|---|---|---|---|
| CCA | $7.7e^2$ | $1.1e^2$ | $2.0e^3$ | $1.8e^3$ | $1.3e^2$ | $9.1e^2$ | $1.2e^3$ | $1.5e^3$ | $8.8e^2$ | $1.1e^2$ | $1.1e^3$ | $2.7e^2$ | $6.7e^2$ |
| KSG | $\mathbf{3.2e^{-2}}$ | $1.5e^1$ | $4.7e^{-2}$ | $1.6e^{-1}$ | $7.7e^1$ | $7.6e^1$ | $\mathbf{4.6e^{-2}}$ | $5.7e^1$ | $7.5e^1$ | $5.2e^1$ | $\mathbf{2.5e^{-1}}$ | $4.6e^1$ | $4.1e^1$ |
| MINE | $1.2e^3$ | $1.8e^3$ | $1.8e^3$ | $2.3e^5$ | $7.8e^4$ | $4.3e^3$ | $2.8e^3$ | $1.8e^3$ | $2.1e^3$ | $6.7e^4$ | $6.9e^4$ | $5.4e^2$ | $3.5e^3$ |
| InfoNCE | $1.5e^1$ | $3.9e^1$ | $1.4e^1$ | $2.0e^1$ | $1.1e^2$ | $1.3e^2$ | $1.3e^1$ | $8.0e^1$ | $1.1e^2$ | $3.2e^2$ | $3.2e^1$ | $7.1e^1$ | $6.9e^1$ |
| DV | $1.4e^{17}$ | $2.1e^{18}$ | $6.2e^{16}$ | $3.9e^{18}$ | $6.5e^{18}$ | $2.3e^{19}$ | $1.5e^{17}$ | $1.8e^{18}$ | $2.3e^{16}$ | $1.6e^{18}$ | $6.0e^{18}$ | $6.0e^{18}$ | $1.6e^{18}$ |
| LMI | $2.1e^{-1}$ | $1.6e^1$ | $1.4e^{-1}$ | $2.7e^{-1}$ | $8.3e^1$ | $8.4e^1$ | $1.9e^{-1}$ | $5.9e^1$ | $8.0e^1$ | $5.4e^1$ | $2.9e^{-1}$ | $4.7e^1$ | $4.2e^1$ |
| MINDE | $8.8e^{-1}$ | $1.4e^1$ | $1.3e^0$ | $9.2e^{-1}$ | $8.2e^1$ | $8.0e^1$ | $6.8e^{-1}$ | $5.7e^1$ | $7.3e^1$ | $5.5e^1$ | $9.6e^{-1}$ | $4.4e^1$ | $3.6e^1$ |
| MIST | $6.5e^{-2}$ | $\mathbf{1.5e^0}$ | $5.4e^{-2}$ | $2.5e^{-1}$ | $\mathbf{1.4e^0}$ | $2.0e^0$ | $1.6e^{-1}$ | $1.6e^0$ | $9.5e^{-1}$ | $2.0e^0$ | $3.4e^{-1}$ | $\mathbf{2.1e^1}$ | $\mathbf{3.0e^1}$ |
| MIST$_{QR}$ | $3.4e^{-2}$ | $\mathbf{1.5e^0}$ | $\mathbf{2.6e^{-2}}$ | $\mathbf{1.4e^{-1}}$ | $1.5e^0$ | $\mathbf{1.6e^0}$ | $9.7e^{-2}$ | $\mathbf{1.4e^0}$ | $\mathbf{6.9e^{-1}}$ | $\mathbf{1.7e^0}$ | $2.7e^{-1}$ | $3.0e^1$ | $3.6e^1$ |

Table 10: Average inference time per sample measured on $\mathcal{M}_{\text{test-extended}}$.

| Method | Time / sample (s) |
|---|---|
| MIST$_{(QR)}$ | 0.000 55 |
| CCA | 0.000 90 |
| KSG | 0.021 |
| LMI | 0.76 |
| InfoNet | 1.64 |
| InfoNCE | 2.01 |
| NWJE | 2.15 |
| DV | 2.20 |
| MINE | 2.75 |
| VCE | 3.93 |
| FMMI | 15.71 |
| MINDE | 122.7 |

Table 11: Average performance metrics for models of different sizes.

| Model | Params (M) | IMD | | | OoMD | | |
|---|---|---|---|---|---|---|---|
| | | MSE | Abs. Bias | Variance | MSE | Abs. Bias | Variance |
| MIST | 5.5 | 0.7508 | 3.0255 | 0.3401 | 7.6679 | 20.2033 | 0.3743 |
| MIST$_{QR}$ | 5.5 | 0.7637 | 0.6853 | 0.2941 | 9.2189 | 2.9803 | 0.3370 |
| MIST$_{medium}$ | 1.0 | 1.6223 | 5.6388 | 0.4427 | 11.2461 | 30.1180 | 0.4765 |
| MIST$_{QR-medium}$ | 1.0 | 2.0265 | 1.1974 | 0.5928 | 10.5160 | 3.1461 | 0.6183 |
| MIST$_{small}$ | 0.25 | 1.5498 | 5.1608 | 0.4851 | 8.9834 | 24.1663 | 0.5342 |
| MIST$_{QR-small}$ | 0.25 | 1.8910 | 1.1645 | 0.5349 | 11.3058 | 3.2777 | 0.5628 |

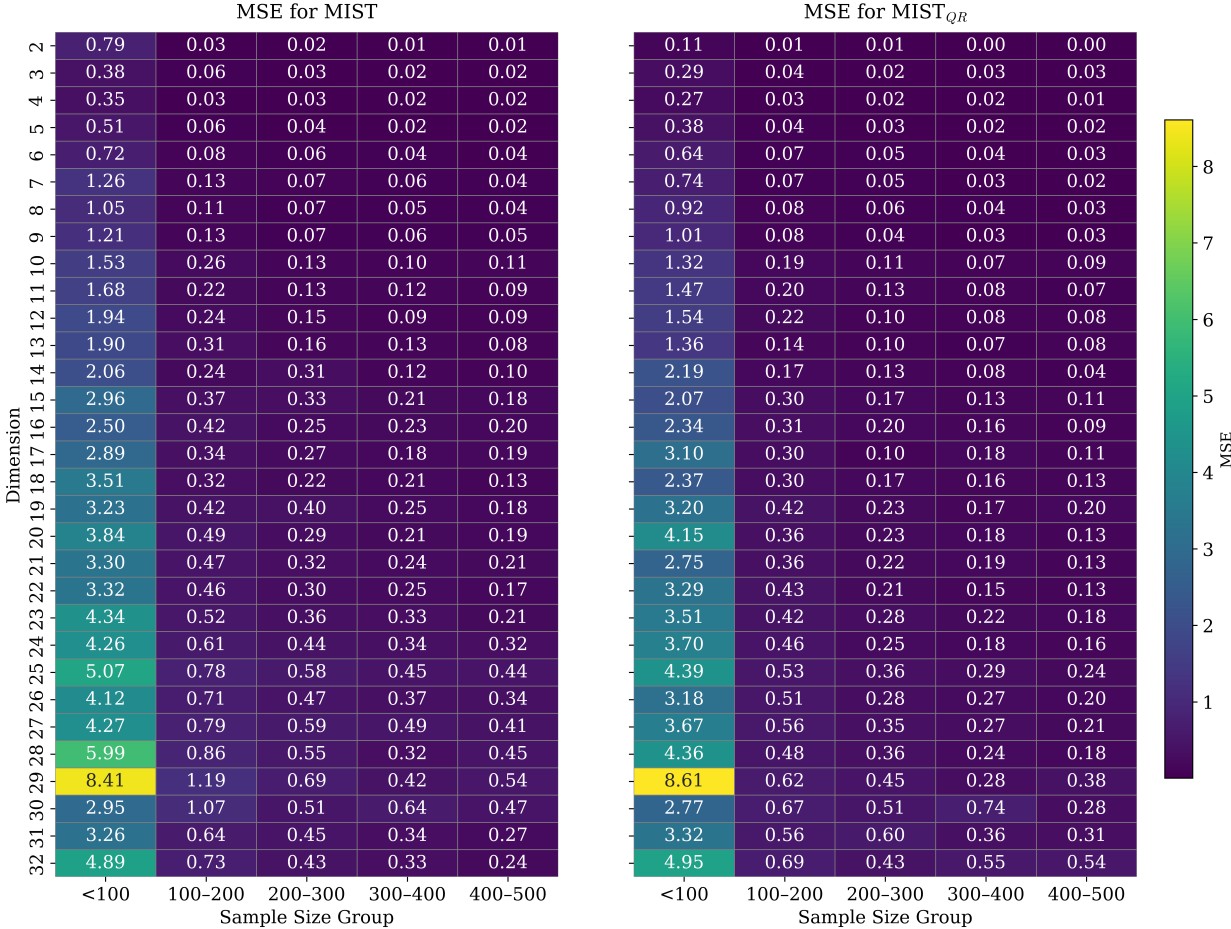

Figure 16: Detailed visualization of the average MSE values for our two proposed models across different sample-size groups and all evaluated dimensionalities. The measurements were obtained on $\mathcal{M}_{\text{test-extended,IMD}}$. The results highlight the increased difficulty of estimating MI in small-sample regimes and the consistent improvement in performance as the sample size grows.

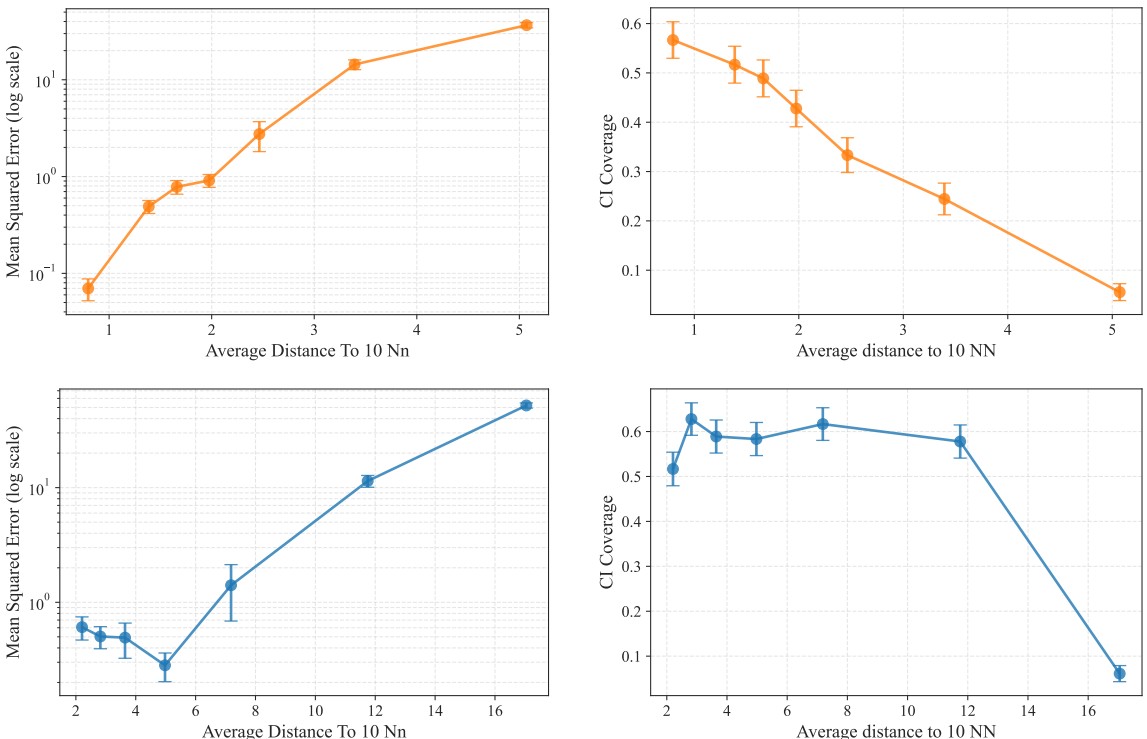

Figure 17: Relationship between distance in MIST embedding space and model performance. Top row: MIST. Bottom row: $\text{MIST}_{QR}$. Left column: mean squared error (MSE) as a function of the average distance to the 10 nearest neighbors in the training embedding space (with K=50,000). Right column: confidence interval (CI) coverage as a function of the same distance. In particular $\text{MIST}_{QR}$ generalizing better stays in higher performance for larger deviation from the training set compared to MIST. For both models, the larger the distance the worse the performance. For MSE the spearman rank correlation (without binning) is 0.68 and for CI coverage it is -0.32.

- At inference time, for each input dataset, we compute its embedding using the same procedure.

- We then identify the $m$ nearest neighbors (m-NN) of this embedding within the reference set and compute the average distance to these neighbors in embedding space using cosine similarity.

- Finally, we analyze how this average m-NN distance correlates with both prediction error (MSE) and confidence interval (CI) coverage.

We report results in Figure 17 using $K = 50{,}000$ and $m = 10$. Overall, we observe that the average m-NN distance is a strong predictor of performance degradation. As the distance to the nearest training embeddings increases, indicating that the input dataset is further out-of-distribution, MIST exhibits a predictable increase in MSE and a decrease in CI coverage. This provides a simple and effective mechanism to assess the reliability of MIST predictions at test time. Interestingly, the ranges of distances is different between MIST and $\text{MIST}_{QR}$ showing that they learn different representational spaces.

We further note that increasing $K$ improves the strength of the correlation, while varying $m$ in the range $[1, 10]$ does not significantly affect the results. We also experimented with alternative distance-based metrics, including distance to the centroid of the reference embeddings and Mahalanobis distance, but found that the average m-NN distance consistently provides the best predictive signal.

In terms of quantitative results, we observe a Spearman rank correlation of approximately 0.68 between the average 10-NN distance and MSE, and $-0.32$ between the average 10-NN distance and CI coverage. We use

rank correlation due to the skewed distribution of MSE values. Most ranking inconsistencies arise in cases where both the distances and performance metrics are very similar, typically corresponding to datasets close to the training distribution where the model performs well.

In terms of computational overhead, recording the embeddings incurs negligible cost. The additional cost of computing the 10-NN over $K = 50,000$ reference embeddings at test time remains small (around 10% increase inference time), such that MIST remains significantly faster than competing baselines and orders of magnitude faster than neural estimators. Moreover, this overhead could be further reduced through the use of optimized nearest neighbor search structures. Overall, the average m-NN distance in MIST embedding space provides a fast, simple, and effective proxy for estimating performance degradation under distribution shift.

## E    Self-Consistency Tests

### E.1    Self-Consistency of Quantile Regression

We checked the consistency of $\text{MIST}_{\text{QR}}$'s quantile predictions in two ways:

- **Monotonicity:** For a given fixed sample $(x, y)$, we sample a random uniform value of $\tau$ in $(0, 1)$ and estimate $\hat{I}_\tau = \text{MIST}_{\text{QR}}(x, y, \tau)$. We repeat this process with $n$ values $(\tau_1, \ldots, \tau_n)$ and check whether $(\hat{I}_{\tau_1}, \ldots, \hat{I}_{\tau_n})$ is an increasing sequence. We perform this operation on each data point of $\mathcal{M}_{\text{test}}$ using $n = 1000$, and count the number of points for which monotonicity is not respected.

- **Bound checking:** For a given fixed sample $(x, y)$ with true MI $I_{\text{true}}$, we compute $\hat{I}_{\text{lower}} = \text{MIST}_{\text{QR}}(x, y, \tau = 0)$ and $\hat{I}_{\text{upper}} = \text{MIST}_{\text{QR}}(x, y, \tau = 1)$. We perform this operation on each data point of $\mathcal{M}_{\text{test}}$, and count the number of points for which $I_{\text{true}} < \hat{I}_{\text{lower}}$ (lower bound failure) or $\hat{I}_{\text{lower}} < I_{\text{true}}$ (upper bound failure).

The results, given in Table 12, show that monotonicity breaks occur relatively rarely on both IMD and OoMD sets (1.1% and 1.4%, respectively). Further analysis reveals that monotonicity failures only occur on points with very low values of $I_{\text{true}} < 0.05$. Bound failures happen at a moderate rate on the IMD dataset, and significantly more often for lower bounds (9.6%) than for upper ones (4.3%). When moving to OoMD distributions, however, those failure rates increase to 15.8% for lower bounds and 27.5% for upper ones, showing that this task remains difficult for our model.

There are several factors that influence the reliability of $\text{MIST}_{QR}$'s quantiles, which we describe below.

First, $\text{MIST}_{QR}$'s quantile distribution is implicitly conditioned on the distribution of target values seen during training. When the distribution of target MI values shifts, the predicted quantile ranges remain anchored to the range and location seen in training, causing systematic undercoverage. The bound failures seen above correspond strongly to location shifts between train and test target distributions. In Table 13, we show the target distribution statistics of each data split. The spread is relatively consistent between train and test splits, but IMD and OoMD have unique location shifts that correspond to the magnitude and asymmetry of bound failures. In IMD, the median is shifted downward moderately, and the observed values in the IMD set are often lower than the values expected by $\text{MIST}_{QR}$. This creates an asymmetric miscalibration, with a higher rate of lower bound failures than upper bound failures. In OoMD, the mean and median both shift substantially upward, with values larger than expected by $\text{MIST}_{QR}$. This creates asymmetry in the opposite direction, with a higher rate of upper bound failures than lower bound failures. Larger location shifts create higher rates of bound failures. Our observations parallel the well-documented challenges in uncertainty estimation under distribution shifts, and we highlight mitigation as an important area of future work.

Second, $\text{MIST}_{QR}$ predictions are influenced by additional sources of variation that occur in meta-prediction. In Figure 5, we can see that $\text{MIST}_{QR}$'s quantiles create a slight S-shaped curve in-distribution, in which the predicted quantile range is slightly narrower than the empirical distribution of observed MIST estimates. This pattern occurs because MIST estimates are influenced by additional sources of variation beyond the

Table 12: Consistency of the $\tau$ parameter of $\text{MIST}_{\text{QR}}$ on the IMD and OoMD subsets of $\mathcal{M}_{\text{test}}$

|  | IMD (1080 points) | OoMD (1260 points) |
|---|---|---|
| Monotonicity breaks | 15 (1.4%) | 12 (1.2%) |
| Lower bound failures | 104 (9.6%) | 199 (15.8%) |
| Upper bound failures | 46 (4.3%) | 346 (27.5%) |

Table 13: Distribution Statistics of Target MI Values across Dataset Splits

| Statistic | Train | Valid | IMD | OoMD |
|---|---|---|---|---|
| Min | 0.000 | 0.000 | 0.000 | 0.000 |
| Q1 | 0.622 | 0.647 | 0.504 | 0.846 |
| Med | 1.398 | 1.371 | 1.028 | 2.363 |
| Q3 | 3.414 | 4.134 | 3.133 | 7.075 |
| Max | 42.818 | 29.721 | 33.577 | 34.591 |
| Mean | 3.271 | 3.262 | 3.175 | 4.644 |
| Std | 4.773 | 4.333 | 5.188 | 5.238 |

sampling uncertainty captured by quantile regression, including model error. These uncertainties increase the variation in MIST predictions, causing the predicted quantiles to be an underestimate of the observed spread of predictions. This undercoverage can increase out of meta-distribution (OoMD), as the distribution shift introduces substantial variation that is not captured by quantile regression.

These limitations reflect foundational challenges in many machine learning and statistical estimation tasks, where re-calibration under distribution shift remains a key challenge and several sources of uncertainty jointly influence prediction quality. We study quantile regression in this work as a useful tool to capture a critical dimension of uncertainty (under limited samples). However, it is important to highlight that quantile regression does not give us a holistic measure of all sources of variation that influence MIST predictions, and may be less reliable under substantial distribution shifts. As a result, careful validation is required.

### E.2 Self-Consistency of the Learned Quantity

Figures 18 through 20 contain the results of three consistency tests applied to MIST and $\text{MIST}_{\text{QR}}$:

- The **independence test** checks that if $X \perp\!\!\!\perp Y$, then $I(X; Y) = 0$.

- The **data processing inequality** checks that if $X, Y$ and $Z$ form a Markov chain $X \to Y \to Z$, then $I(X; Y) \geq I(X; Z)$ (further processing of the data cannot add new information).

- The **additivity over independent samples** test checks that if $(X_1, X_2) \perp\!\!\!\perp (Y_1, Y_2)$, then $I(X_1, Y_1; X_1, Y_2) = I(X_1; X_2) + I(Y_1; Y_2)$

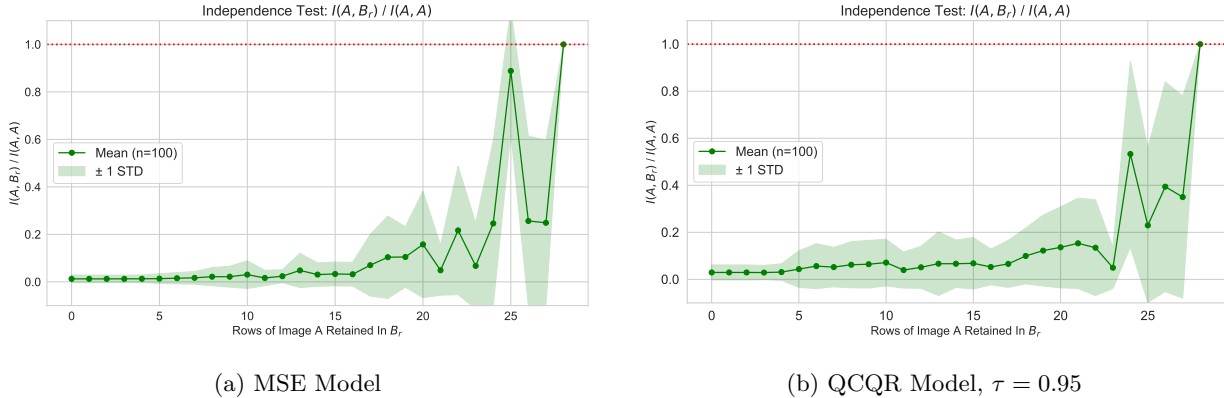

(a) MSE Model

(b) QCQR Model, $\tau = 0.95$

Figure 18: Independence Test: $A$ is an MNIST image, and $B_r$ contains the top $r$ rows of image $A$. For a consistent estimator, the plotted MI ratio will be non-decreasing and approach 1 with an increasing number of shared rows.

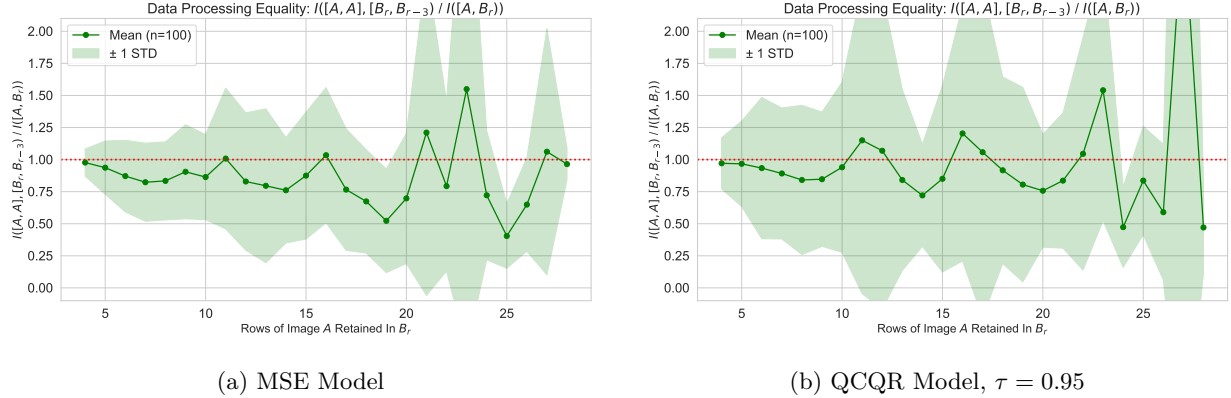

(a) MSE Model

(b) QCQR Model, $\tau = 0.95$

Figure 19: Data Processing Test: $A$ is an MNIST image, and $B_r$ contains the top $r$ rows of image $A$. This test compares $I([A,A],[B_r,B'_{r-3'}])$ to $I([A,B_r])$, where $A$ is an original MNIST image, $B_r$ is the first $r$ rows of $A$, and so $B'_{r-3'}$ is a further masked version of image $A$. The test produces a ratio which should be near to 1 for all values of $r$, which amounts to checking that adding a further masked version of your image $B_r$ does not change the predicted MI. Images are encoded with PCA fit on 10K random samples, and error bars of $1\sigma$ reported, with 100 bootstrapped samples.

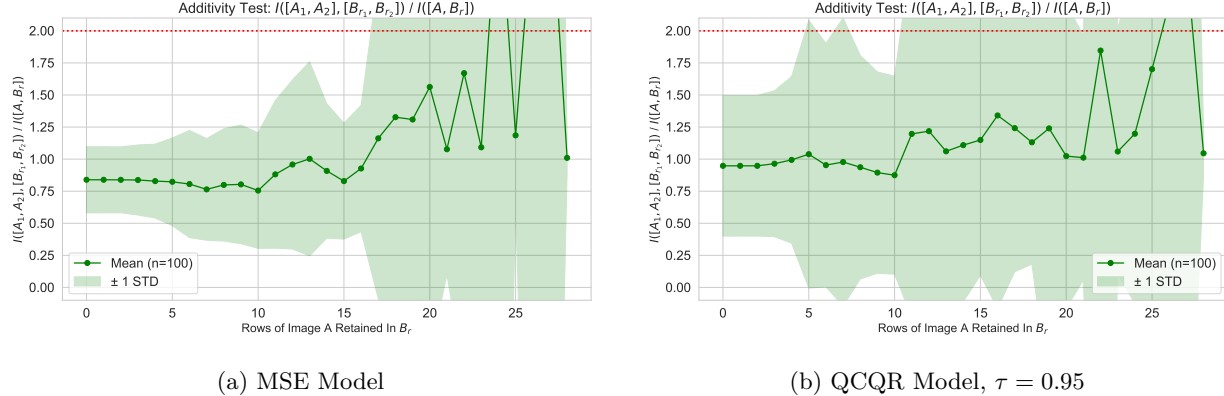

(a) MSE Model

(b) QCQR Model, $\tau = 0.95$

Figure 20: Additivity Test: $A$ is an MNIST image, and $B_r$ contain the top $r$ rows of image $A$. The premise of this test is that adding a second independent sample should double the mutual information estimate. The ratio compares $I([A_1, A_2], [B_1 r, B_2 r])$ to $I([A, B_r])$, and should be near 2 for all values of $r$. Images are encoded with PCA fit on 10K random training samples. Error bars of $1\sigma$ are reported with 100 bootstrapped samples.

## F  Feature Selection

In this experiment, we explored using our estimators in two synthetic feature learning settings. We define $X \in \mathbb{R}^{n \times d}$ as a random normal matrix with $d$ features and $n$ samples. Next, we define a set of weights for these features, each bounded between 0 and 1, $w \in \{0, 1\}^{d \times 1}$. The label Y is given by $Y_{\text{true}} = Xw^* + \epsilon$, where $Y \in \mathbb{R}^{n \times 1}$ is the matrix multiplication of $X$ and pre-set masking weights $w^*$ plus added normal noise $\epsilon$. The goal of this task is to recover the true weights $(w^*)$, which correctly assign a value of 1 for the two informative features and 0 otherwise. In both settings, we set the mean and variance of $X$ to 0 and 1.0 respectively, and the normal noise applied to $Y_{true}$ has 0 mean and variance of either 0.3, 0.5, or 1.0. We generate 32 features and randomly select 2 features to be informative.

**Direct Feature Selection**  In *Direct Feature Selection*, we use the mutual information between each feature and the outcome as the feature score. We train a new MINE estimator for 200 steps for each prediction, and follow the parameters in MINE's repository, using Adam, a learning rate of 0.001, the biased loss, and a critic hidden dimension of 64. For KSG, we use K=5 for sample sizes of 10 or greater, and use K=2 for sample size 5. In Figure 22, we plot the ROC curves of each estimator by noise level, and in Figure 24, we provide a plot of the feature estimates for a random seed of data when 50 samples were provided and the noise was set to 1.0.

**Learned Feature Selection**  For *Learned Feature Selection* setting, we learn a continuous set of feature weights using gradient ascent. For *Learned Feature Selection*, we use the MSE-trained MIST, KSG and MINE estimators., For MINE and MIST, perform gradient ascent with the Adam optimizer and learning rate of 0.01 for 1000 steps. For MINE, we consider two optimization settings. In the first, we continually train one MINE estimator, allowing 15 steps of updates with each prediction. In the second setting, we train a new MINE estimator for 200 steps for each prediction. In both cases, we follow the parameters in MINE's repository, using Adam for the optimizer and a learning rate of 0.001. We used the biased loss and set the critic hidden dimension to 64. Weights are initialized with a small constant and bounded between 0 and 1 for all estimators. For a candidate set of weights, the predicted label is $Y' = Xw' \in \mathbb{R}^{n \times 1}$, produced by the candidate weights $w'$. The best weights are chosen as $w_{\text{best}} = \arg\max_{w'} \text{MIST}(Y_{\text{true}}, Y')$, which maximizes the mutual information with the true label.

In Figure 21, we plot the ROC curves of the learned features by noise level, with 20 random seeds each. We find that even with a very limited number of samples, MIST estimates provide an accurate and stable learning signal. Training MINE to convergence at each prediction achieves better performance than continually MINE

training, but remains significantly less effective than using MIST estimates. In Figure 23, we plot the top-k accuracy according to the number of samples provided. Accuracy is reported as the proportion of the two highest weighted features that are truly informative. While the number of informative features is rarely known ahead of time, we provide this view of the results to better compare the method performance at each sample size. We find that MIST-guided selection is more accurate and lower variance on average, achieving perfect accuracy at 100 samples. In Figure 25, we show the feature weights through training of two randomly chosen seeds with noise variance set to 1.0.

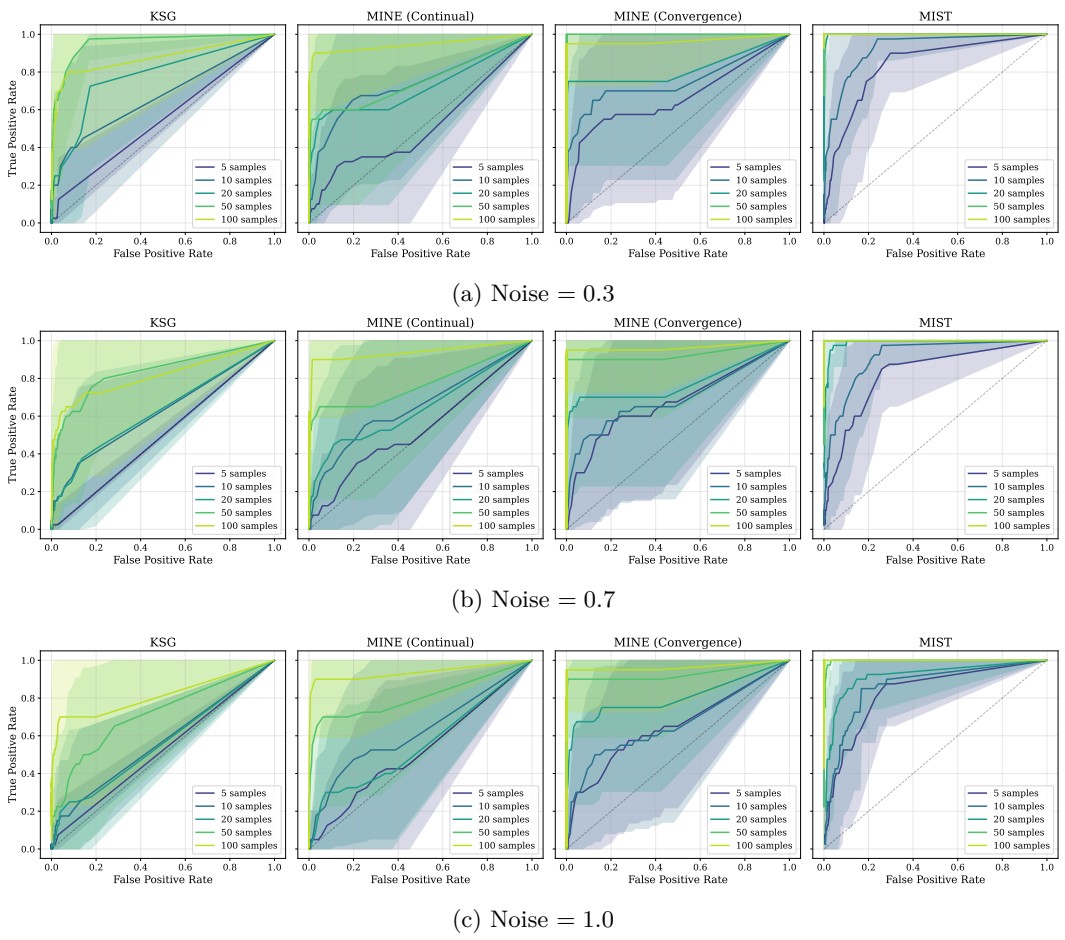

(a) Noise = 0.3

(b) Noise = 0.7

(c) Noise = 1.0

Figure 21: ROC curves for learned feature selection across noise levels, estimated over 20 random seeds with $\pm 1$ standard deviation shaded. MIST-guided optimization is more accurate and lower variance than MINE across all settings.

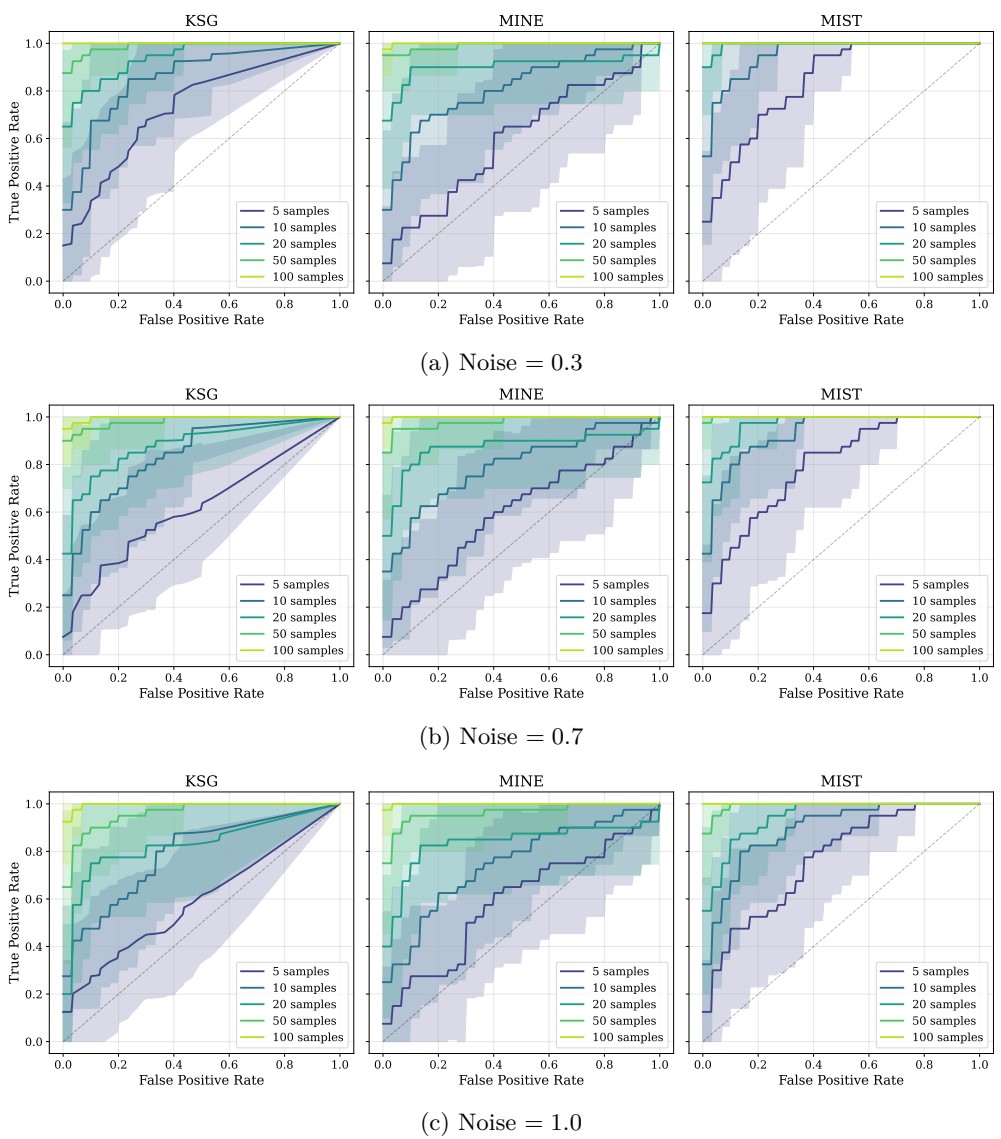

(a) Noise = 0.3

(b) Noise = 0.7

(c) Noise = 1.0

Figure 22: ROC curves for direct feature selection across noise levels, estimated over 20 random seeds with ±1 standard deviation shaded.

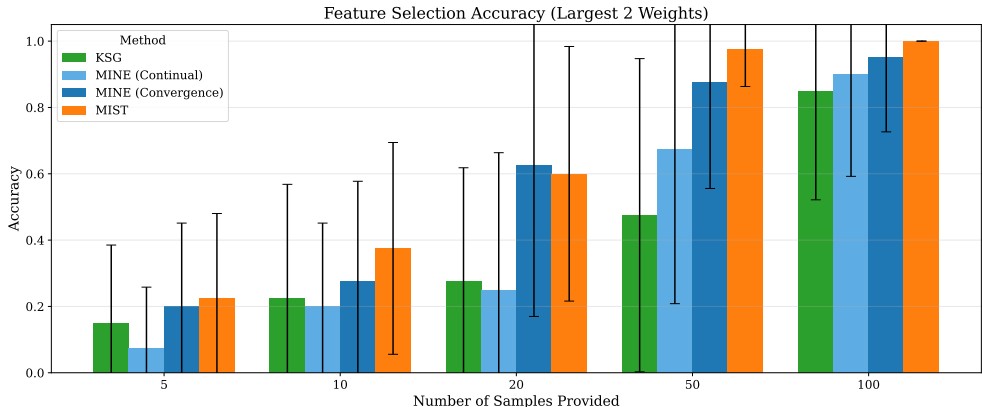

Figure 23: Comparison of the accuracy of features selected via gradient ascent on MINE predictions (either trained continually and trained to convergence) and MIST predictions. Accuracy is calculated as the proportion of the highest two weights which are the informative features. In other words, how often are top 2 weights assigned to the two correlated features? We find that MIST-guided feature selection is more accurate and lower variance.

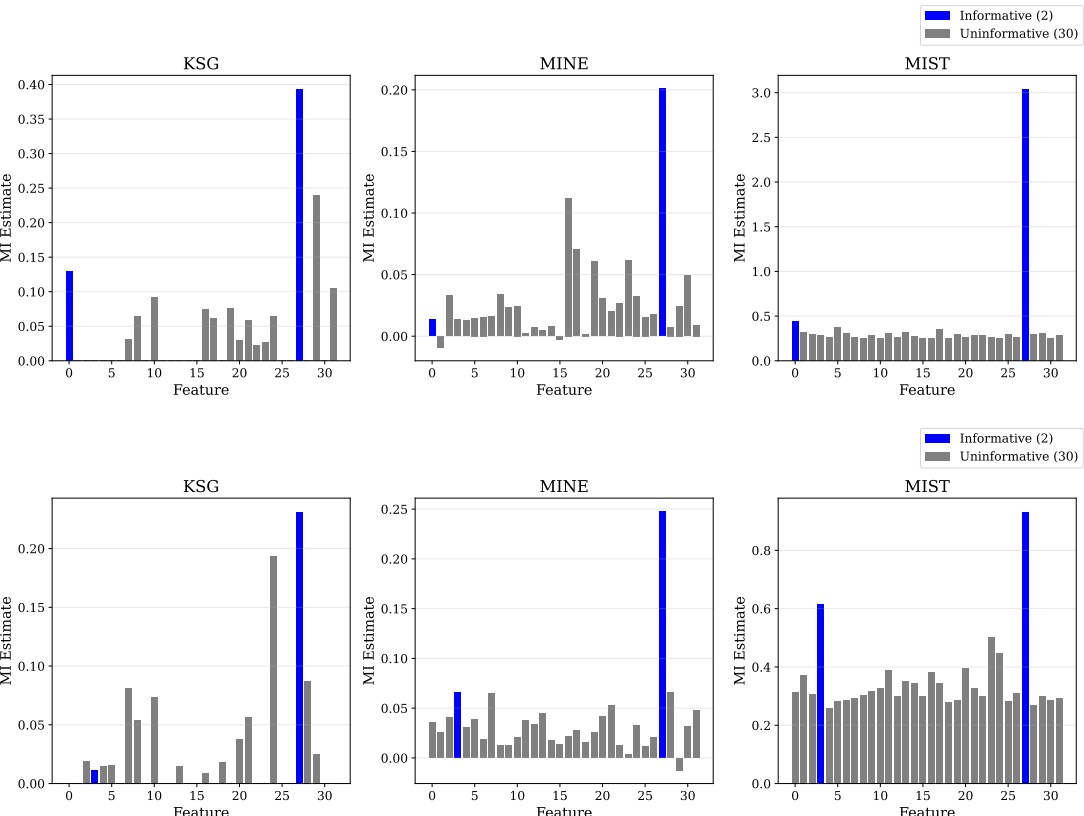

Figure 24: Direct feature selection using KSG, MINE, and MIST (columns). Features are scored by their estimated estimated mutual information with the outcome variable $Y$. Results shown for two randomly selected seeds (rows).

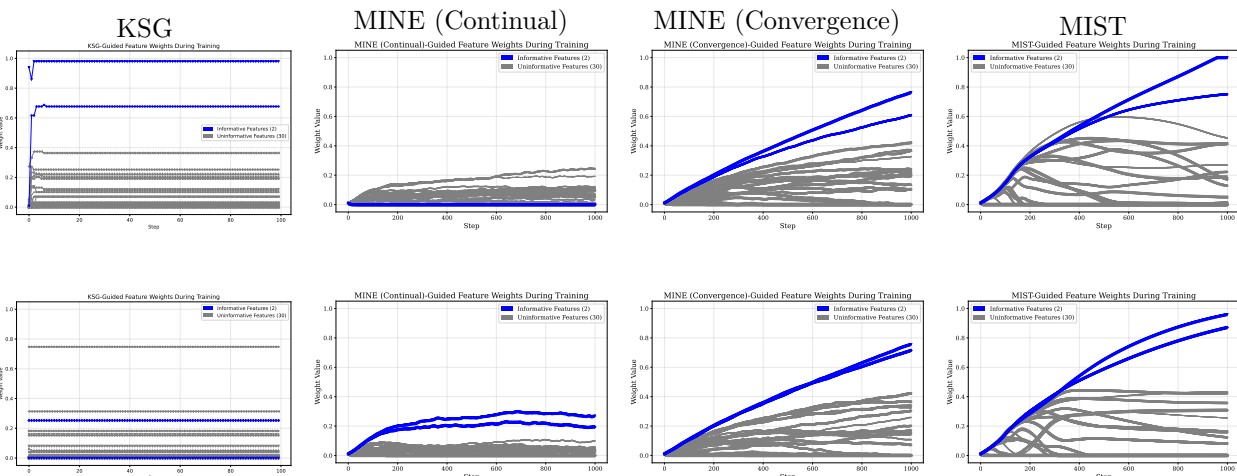

Figure 25: Comparison of feature selection guided by KSG (leftmost), MINE with continual training, MINE with convergence training, and MIST (rightmost). Plotted are the feature weights throughout optimization on two randomly selected seeds (rows). MIST-guided optimization most reliably differentiates informative from uninformative features.

