# OpenReview forum: "MIST: Mutual Information Estimation via Supervised Training"
_TMLR — Accepted by TMLR_

### Review · Reviewer_9siY · 2026-03-16

**Summary Of Contributions:**

The paper proposes MIST, a learned estimator that predicts mutual information (MI) directly from paired samples using supervised training on a large synthetic meta-dataset. The estimator is implemented with a SetTransformer-style architecture extended with attention over both samples and feature dimensions to accommodate variable sample sizes and dimensions. The authors also introduce a quantile-regression variant to approximate the sampling distribution and provide uncertainty estimates via pinball loss. Experiments on synthetic benchmarks show lower MSE than baselines. The paper also discusses calibration, efficiency, and downstream use in feature selection.

**Strengths**

- The paper introduces a permutation-aware neural estimator based on SetTransformer++ with an additional attention mechanism over feature dimensions so that one model can process varying sample sizes and dimensions.
- The quantile-regression version trained with pinball loss to provide uncertainty estimates and approximate the sampling distribution of MI.
- It reports broad empirical comparisons showing lower MSE than many baselines on the proposed benchmark, along with analyses of bias, variance, calibration, scaling with dimension/sample size, generalization, and inference speed.

**Weaknesses**

- Claims about a “new paradigm,” “general-purpose” estimators, or adaptability to arbitrary modalities via normalizing flows lack supporting evidence. The paper does not empirically validate modality transfer or fine-tuning on real non-synthetic modalities in the main experiments.
- The paper emphasizes performance on “unseen distributions,” but OoMD families remain generated within the BMI-based synthetic pipeline rather than representing independent benchmark sources. The feature-selection example appears only in Appendix E and primarily compares with MINE-based optimization, rather than a broader set of downstream alternatives.
- Equation numbering appears inconsistent. The introduction labels the MI definition as Eq. (1), while Sec. 3 reuses Eq. (1) for the MSE objective. Appendix A then restarts numbering again at Eq. (4).

**Audience:**

Yes

**Audience Explanation:**

Researchers interested in mutual information estimation, neural estimation of statistical quantities, amortized inference, and uncertainty-aware estimators would likely find the paper informative.

**Claims And Evidence:**

No

**Claims Explanation:**

The central empirical claim is that MIST and MISTQR achieve lower error than several baselines on the authors’ synthetic benchmark, especially in low-sample regimes. This is supported by substantial evidence in Section 4 and Appendix C.

The main gap is between the strong framing claims and the presented evidence. Claims about broad adaptability to arbitrary modalities, foundational/general-purpose use, and embedding in larger pipelines are only partially supported. The paper provides rationale and one downstream appendix example, but not a broad empirical demonstration across modalities or real-world tasks. Those claims would be better presented as promising directions unless additional evidence is added.

**Requested Changes:**

- Temper or qualify claims about “general-purpose,” “foundational,” and arbitrary-modality adaptation unless supported by additional experiments, and explicitly distinguish demonstrated results from future directions.
- Clarify that “unseen distributions” refers to unseen families within the BMI-generated synthetic benchmark. Either broaden Appendix E with stronger downstream comparisons or narrow the related claims in the main text to reflect the limited evidence currently shown.

---

> ### Author Response · Authors · 2026-04-02
> **Tempering Framing and Clarifying Unseen Distributions**
>
> Thank you for your careful review of our work, and useful suggestions.
>
> **Tempering Framing**:
> We have revised the abstract, introduction, and discussion to narrow our framing and clearly separate our discussion of our paper’s claims and promising directions of future work.
> Specifically, we have:
> - Removed descriptions of future work (adaptation to different modalities, embedding in existing pipelines) from the abstract and introduction.
> - Rephrased descriptions of meta-learned statistical estimators as a ‘new approach’ rather than a ‘new paradigm’.
> - Removed instances of ‘foundational’/ ‘general-purpose’ and replaced them with more specific descriptions, such as MIST not requiring test-time optimization.
> - Clearly specified future work in the discussion section
>
> **Clarifying OoMD and Improving Downstream Experiments**:
> We also appreciated your suggestion to clarify that the unseen distributions represented by the OoMD dataset are still generated by the BMI framework. We have added clarifying statements in our description of the OoMD dataset throughout the paper (e.g., ‘unseen distributions ‘ => ‘unseen distributions in our framework’, ‘unseen BMI distributions’). We have also added a separate limitations section (Section 5) in which we discuss the limitations of MIST’s performance out of meta-distribution, and practical recommendations for users, including a lightweight diagnostic tool to detect very out-of-distribution settings.
>
> For the downstream evaluation, we have expanded the feature selection experiment to include KSG as baseline, and added a complementary setting of direct feature selection (in addition to learned feature selection). We find that MIST outperforms MINE and KSG in both settings, and have written a more thorough description of these results in a new results section (Section 4.8).
>
> **Equation Numbering**:
> Thank you for catching this, we have corrected it.

---

### Review · Reviewer_kMQQ · 2026-03-19

**Summary Of Contributions:**

This paper proposes MIST (Mutual Information estimation via Supervised Training), a meta-learning approach to MI estimation. Rather than deriving an estimator from density or density-ratio formulations, MIST parameterizes the estimator as a neural network trained end-to-end on a large synthetic meta-dataset (~625,000 joint distributions with known ground-truth MI values). A SetTransformer++ backbone handles variable sample sizes and dimensions. A companion model, MIST_QR, uses quantile regression to produce calibrated uncertainty intervals without bootstrapping. The paper includes comparisons against ten baselines across 23 distribution families, sample sizes from 10 to 500, and dimensions from 2 to 32.

**Key strengths:** The meta-learning paradigm is a genuine conceptual contribution, clearly motivated and well-contextualized relative to prior work. The empirical scope substantially exceeds most prior evaluations. MIST_QR's built-in calibration is practically valuable, and inference is orders of magnitude faster than neural baselines. Generalization to out-of-meta-distribution (OoMD) families is meaningfully demonstrated.

**Key weaknesses:** MIST does not carry theoretical self-consistency guarantees (data processing inequality, additivity), and the practical significance of these failures is underexplored. The meta-distribution covers only three base families. MIST_QR bound failure rates on OoMD data (up to 27.5%, Table 9) are non-trivial but receive limited attention in the main text. The sole downstream evaluation (feature selection, Appendix E) compares only against MINE.

**Audience:**

Yes

**Audience Explanation:**

MI estimation underpins a wide range of methods in representation learning, feature selection, causal inference, and information-theoretic analyses of networks. The finding that a meta-learned estimator outperforms both classical and neural baselines in low-sample, moderate-to-high-dimensional settings, and that it offers built-in, fast uncertainty quantification, is of direct practical relevance to this audience.

**Broader Impact Concerns:**

This work raises no significant ethical concerns. MI estimation is a general statistical tool, and its methodology is a contribution to the research community with no obvious pathways to harm.

**Claims And Evidence:**

Yes

**Claims Explanation:**

The core claims are all well-supported by Table 1, Figures 3–5, and Table 7. Two caveats apply: the self-consistency analysis in Appendix D reveals bound failures ranging from 4.3% to 27.5% of test points (Table 9), which the main text treats briefly; and the feature selection experiment compares MIST only against MINE, limiting the support for claims about downstream utility.

**Requested Changes:**

**[Critical] Discuss MIST_QR bound failure rates more prominently.**
Table 9 shows lower/upper bound failure rates of 15.8%/27.5% on OoMD data, yet these numbers appear only in Appendix D. Given that calibrated uncertainty is a headline contribution, the authors should summarize these results in Section 4.4 or 5, and discuss which distributional or sample-size conditions tend to produce failures.

**[Critical] Clarify generalization limits more precisely.**
Section 4.5 and Figure 6 show that MIST struggles when both the distribution family and dimensionality are unseen simultaneously. The paper should be more explicit about what users need to know about their data to trust MIST's outputs, and whether any practical diagnostic exists for detecting this failure mode.

**[Recommended] Broaden the downstream evaluation.**
The feature selection experiment (Appendix E) compares MIST only against MINE. Including at least one additional baseline (e.g., KSG with gradient-free optimization) would better support claims about MIST's utility in optimization pipelines.

**[Recommended] Discuss the meta-distribution design more critically.**
The training set covers only three base families (multivariate normal, Student-t, and uniform additive noise) with a fixed set of invertible transformations. A brief discussion of which real-world distributions are likely to be poorly covered (e.g., discrete marginals, multimodal densities, heavily structured data) would help practitioners assess the scope.

**[Recommended] Clarify the InfoNet comparison.**
InfoNet is the most directly comparable prior work, but the experimental comparison is limited: it appears only in Table 1 on the smaller M_test set. A brief discussion of how MIST and InfoNet differ in OoMD generalization would be valuable on this key axis.

**[Recommended] Report median loss alongside mean in Table 1.**
Given the very large standard deviations for some baselines (e.g., MINE), reporting median MSE alongside the mean would give a clearer picture of typical vs. worst-case behavior.

**[Minor] Discuss VCE's high failure rate.**
VCE failed to produce estimates for 99/216 (IMD) and 118/252 (OoMD) test points (Table 1 footnote). This deserves at least a sentence of discussion rather than a footnote.

---

> ### Author Response · Authors · 2026-04-02
> **Discussion of Calibration Out of Meta Distribution, Generalization Limits, and Broadening Downstream Evaluation**
>
> Thank you for your thoughtful and detailed review! We appreciate that you found our work to be a genuine conceptual contribution that is well motivated and contextualized with prior work. We are also encouraged that you found our experimental scope to be substantially exceeding prior work and findings that are of direct practical relevance to the TMLR audience. Thank you for your insights on how to improve our work, which we found quite helpful. We respond to each below:
>
> **More Detailed Discussion of Calibration Degradation Out-of-Distribution**:
> Thank you for the suggestion to explore this limitation in greater detail. We have done so in a few ways:
> First, we have added a limitations section with the goal of more clearly and thoroughly discussing MIST’s limitations and potential failure modes for users, including the bound failures rate in the OoMD setting.  In this section (5.2), we include a summary of the results on the calibration consistency and degradation OoMD that we studied in Appendix E (previously Table 9, now Table 12).
> Second, we have expanded our analysis and discussion in Appendix E, including a more detailed description of what MIST_QR’s quantiles capture, the failure patterns we observe, and the conditions under which they fail as a bound.
> Third, we have added a lightweight, simple diagnostic tool based on MIST embeddings to detect if a sample is very OoMD (mentioned in ‘Mitigations’ in Section 5, and described in detail in Appendix D), and find embedding distances to be highly correlated with performance degradation in both accuracy and confidence interval coverage. We believe this approach can be of practical use to help users determine if MIST’s predictions and quantile intervals are reliable for their data.
>
> **More Explicit Discussion of Generalization Limits**:
> Thank you also for this suggestion, which we found quite helpful! In the new limitation section (Section 5), we have provided an explicit discussion of when MIST is expected to be reliable, and under what axes we have found it to generalize well or not. We also describe the mitigations we suggest for practitioners, including the OoMD detection tool described above.
>
> **Additional Baselines for Downstream Evaluation**:
> As suggested, we have added KSG as a baseline for the learned feature selection experiment (previously Appendix E, now Sec 4.8 and Appendix F), using coordinate descent. We find that MIST substantially outperforms KSG as well as MINE. We further expand the downstream evaluation with a complementary setting, in which features are selected directly by their shared information with the outcome (MI ($X_i$;Y)), rather than learning continuous feature weights under optimization. This is a simpler task for estimators and allows for a comparison with KSG without optimization. We find that MIST outperforms MINE and KSG in this direct feature selection setting as well, and have written a more thorough description of these results in a new results section (section 4.8).
>
> **Discuss Meta-Distribution Design**:
> We thank the reviewer for this suggestion, and have added example distributions that are unlikely to be in meta-distribution for MIST to the limitations section.

---

> ### Author Response · Authors · 2026-04-02
> **Clarification of InfoNet Comparison, Addition of Median Results, and Discussion of VCE Failure Rate**
>
> **Clarify InfoNet Comparison**:
> To better compare with InfoNet, we have made the following changes, all in Appendix C.3. First, we add discussion of our OoMD comparisons between MIST and InfoNet. The OoMD results in Table 1 represent data that is out-of-distribution for both methods. However, InfoNet was trained and evaluated on only multivariate gaussian distributions, motivating the additions below.
>
> To provide a more thorough and fair comparison between InfoNet and MIST, we added a separate evaluation experiment in Appendix C.3. which compares MIST and InfoNet on the portion of the test set which we expect to be in-distribution for InfoNet (multivariate gaussian distributions). In Table 7, we report the results on the multivariate gaussian distributions directly, without any transforms applied. This data is well in-distribution for both models. We find that MIST largely outperforms InfoNet on these distributions in average and median loss. For the lowest sample setting, we find InfoNet has slightly lower mean loss but a higher median loss than MIST. In all other sample settings, MIST outperforms InfoNet substantially.
>
> In Table 8, we report the results on the portion of the test set which includes multivariate gaussian distributions under invertible transforms. This allows us to compare InfoNet and MIST on more complicated data that still should be approachable by both methods, as the base distribution is still a multivariate gaussian. This set includes transformations seen by MIST during training (IMD) and also transforms that have never been seen by MIST (OoMD). We see that MIST substantially outperforms InfoNet across sample sizes on both the transforms seen during training (IMD) and those unseen in training (OoMD).
>
> Our results across these settings indicate that MIST achieves largely better performance on data that is in-distribution for both MIST and InfoNet, and generalizes much better to transformed data (Table 8), as well as larger sets of complex data (Table 1, OoMD).  Finally, despite also being amortized, InfoNet remains orders of magnitude slower to run at inference time compared to MIST (Figure 7).
>
> **Report Median Values in Table 1**:
> We report the median values of our main results in Table 6 (Appendix C.2), and find that MIST outperforms baselines in median loss as well as the mean, with KSG achieving slightly lower median loss for one setting: lowest sample size IMD data. Therefore, as the reviewer expected, we observe a highly skewed error profile for most baselines.
>
> **Discuss VCE’s Failure Rate**:
> We agreed with this point, and added a more detailed technical description of the failures we found with VCE in the caption of Table 1. (The very low sample regimes create numerical issues when inverting covariance matrices.)

---

### Review · Reviewer_Gr63 · 2026-03-21

**Summary Of Contributions:**

This paper proposes (MIST), a supervised meta-learning approach to estimate mutual information, which directly predicts MI from a given sample set, rather than estimating densities or density ratios first. Such an estimator is trained end-to-end on a large synthetic metal dataset constructed by joint distributions with known MI. The estimator adopts a known SetTransformer-style architecture with an added attention mechanism over feature dimensions, and a quantile regression variant (MISTOR) for uncertainty estimates. Particularly, the paper studies difficult regimes that prior MI works fail at (small sample size, high dimension). In numerical experiments, the paper presents results that show strong gains over classical baselines.

**Audience:**

Yes

**Audience Explanation:**

The estimation of mutual information is important in many research problems, for example, the information-theoretic study of machine learning. The contribution is of practical interest.

**Claims And Evidence:**

Yes

**Claims Explanation:**

The claims are supported by numerical evidence,

**Requested Changes:**

One concern is that MIST is a good estimator learned from large synthetic tasks generated by BMI, which limits the prediction of MIST to those seen in the training. This is also acknowledged by the authors. Can the authors quantify how performance degrades when true MI exceeds the training range, or when samplee sizes are much larger than those seen in training?

---

> ### Author Response · Authors · 2026-04-02
> **Generalization to Larger Sample Sizes and True MI**
>
> Thank you for your thoughtful review. We are glad that you found our work to be tackling a problem important in many research areas, and that our contributions are relevant and supported. We agree that MIST’s performance is inherently dependent on the training distribution, and it is essential to acknowledge and study this in our work. We address both points below:
>
> **Generalization to Larger Sample Sizes**:
> In Section 4.5, we evaluated MIST’s generalization to larger sample sizes on a version of MIST trained on a restricted training set. In this experiment, we train on examples with up to 300 samples and evaluate up to 500 samples, and evaluate the sample generalization under IMD and OoMD settings and under increasing dimensionality (Fig 6). Our results demonstrate that MIST’s performance continues to improve with more samples, even beyond the sample sizes seen in training. We believe this experiment addresses your request to evaluate performance at sample sizes much larger than those seen during training. Please let us know if there are any remaining concerns.
>
> **Generalization to Larger True MI**:
> Because MIST learns to predict MI on a range of label values in the train distribution, it is not expected to predict greater than the maximum value seen during training. We observed this in the feature learning experiment, where estimates plateaued near the maximum label value of the train set. Our training set covers a much larger range of values than estimators are conventionally evaluated (up to ~40, while most MI estimators are evaluated only for MI less than 5, See Appendix A.3 Table 3). Therefore, we believe MIST’s prediction range will have significant coverage over use cases. However, this remains a key limitation that is important to highlight for practitioners, and we do so explicitly in a new limitations section (Section 5).

---

### Author Response · Authors · 2026-04-02
**Summary of Author Response**

We thank the reviewers for their thoughtful engagement with our work, and insightful suggestions to improve it. We were encouraged to see consensus among reviewers on the importance of MI estimation across research domains, the rigor of our empirical scope, and the practical relevance of our contributions for the TMLR audience. Below is a summary of the main concerns raised, and how we have addressed them:

**Understanding Limitations of the Meta-Distribution**:
 Reviewers asked about MIST’s generalization beyond the training meta-distribution, and how users would know when MIST is expected to be reliable. In response, we created a separate limitations section (Section 5) that summarizes our experimental findings for practitioners, including the axes on which MIST generalizes well and where we find performance and calibration degrades. We also introduce a simple OoMD diagnostic tool based on MIST’s representations, which we find to be highly correlated with performance and calibration degradation.

**Expanding Downstream Evaluation**:
 Reviewers asked for KSG to be added to the feature selection experiment. We have added KSG, and introduced a complementary setting in which features are evaluated independently rather than learning continuous weights. Both are reported in a new section (4.8), and MIST outperforms baselines in both settings.

**Tempering Framing and Separating Results From Future Directions**:
One reviewer suggested narrowing the framing by better separating our results from future directions (specifically, claims of modality transfer, utility in other pipelines, and general-purpose, foundational phrasing). We have revised phrasing in the abstract and introduction accordingly, and discuss the future directions in the discussion.

---

### Decision · Action_Editor_U8Cm · 2026-04-27

**Recommendation:** Accept as is

**Audience:**

Yes

**Audience Explanation:**

Mutual information is a wide-reaching statistical quantity that is ubiquitous in modeling, training, and data analysis. Proposals for general-purpose techniques to achieve better and more practical estimation of MI (coupled with public code) are assuredly going to have an audience in the TMLR community.

**Claims And Evidence:**

Yes

**Claims Explanation:**

The goals of this paper are clear; the authors are interested in flexible and reliable estimation of mutual information (prioritized over formal guarantees). The substantive contribution of this work is a neural network based approach proposal, and rigorous empirical evaluations of their proposed technique. All the reviewers found the evidence sufficiently convincing, and the paper itself is well-written and sufficiently clear; the main claims are solid.